# Gene editing and scalable functional genomic screening in *Leishmania* species using the CRISPR/Cas9 cytosine base editor toolbox LeishBASEedit

**Markus Engstler, Tom Beneke***

Department of Cell and Developmental Biology, Biocentre, University of Würzburg, Wuerzburg, Germany

**Abstract** CRISPR/Cas9 gene editing has revolutionised loss-of-function experiments in *Leishmania*, the causative agent of leishmaniasis. As *Leishmania* lack a functional non-homologous DNA end joining pathway however, obtaining null mutants typically requires additional donor DNA, selection of drug resistance-associated edits or time-consuming isolation of clones. Genome-wide loss-of-function screens across different conditions and across multiple *Leishmania* species are therefore unfeasible at present. Here, we report a CRISPR/Cas9 cytosine base editor (CBE) toolbox that overcomes these limitations. We employed CBEs in *Leishmania* to introduce STOP codons by converting cytosine into thymine and created http://www.leishbaseedit.net/ for CBE primer design in kinetoplastids. Through reporter assays and by targeting single- and multi-copy genes in *L. mexicana*, *L. major*, *L. donovani*, and *L. infantum*, we demonstrate how this tool can efficiently generate functional null mutants by expressing just one single-guide RNA, reaching up to 100% editing rate in non-clonal populations. We then generated a *Leishmania*-optimised CBE and successfully targeted an essential gene in a plasmid library delivered loss-of-function screen in *L. mexicana*. Since our method does not require DNA double-strand breaks, homologous recombination, donor DNA, or isolation of clones, we believe that this enables for the first time functional genetic screens in *Leishmania* via delivery of plasmid libraries.

*For correspondence:
tom.beneke@uni-wuerzburg.de

**Competing interest:** The authors declare that no competing interests exist.

## Editor's evaluation

Leishmanias cause a wide spectrum of diseases in many animals: in humans, symptoms range from disfiguring skin lesions to lethal visceral infections. Knowledge of gene function has been limited by aneuploidy, as well as the absence of RNAi (in most but not all species) and of non-homologous DNA end joining (which limits the utility of Crispr-Cas systems). In this paper, the authors describe the successful testing and application of a new method for functional screens in Leishmania, in which targeted cytosine base editing is used to introduce premature stop codons within open reading frames. The advantages and disadvantages of this approach, in comparison with others, are described in a balanced and careful way. This is a very important new tool for researchers in Leishmania, and might also serve as a model for other poorly tractable species.

## Introduction

Leishmaniasis is a neglected tropical disease with poor treatment options, causing an estimated 1 million new cases and 20,000 deaths annually (*World-Health-Organization, 2023*). It is caused by *Leishmania* species, which proliferate in sand flies as motile promastigotes and in vertebrates

as immotile amastigotes. When injected into a mammalian host, promastigotes predominantly enter macrophages (*van Zandbergen et al., 2004*; *Chaves et al., 2020*) and differentiate inside the phagolysosome into amastigotes. In the host, parasites can remain local (cutaneous leishmaniasis), disseminate throughout the skin (diffuse cutaneous/mucocutaneous leishmaniasis), or enter the bloodstream to establish visceral infections (visceral leishmaniasis). These outcomes mark the difference between a mild illness and a deadly disease (*McCall et al., 2013*), and a systematic large-scale genetic dissection of *Leishmania* genes involved in the mechanisms that drive these different clinical manifestations remains to be performed.

Despite this, large-scale genetic screens represent a major challenge for the *Leishmania* research community. While genome-wide RNA interference (RNAi) screens have enabled a detailed functional dissection of nearly all genes in the closely related parasite *Trypanosoma brucei* (*Alsford et al., 2011*; *Horn, 2022*; *Morris et al., 2002*), this success has not been mirrored in *Leishmania*. Recent developments for the use of RNAi in *L. braziliensis* are promising and may facilitate genome-wide loss-of-function screens in the future (*de Paiva et al., 2015*; *Lye et al., 2022*). However, only *Leishmania* species of the *Viannia* subgenus have retained the components required for RNAi activity, precluding its use as a tool in most other *Leishmania* parasites. These include the causative agents of cutaneous and visceral leishmaniasis, such as *L. mexicana*, *L. major*, *L. donovani*, and *L. infantum* (*Lye et al., 2010*; *Ullu et al., 2004*).

An alternative approach to RNAi is the use of CRISPR/Cas9 gene editing. By simplifying previous labour-intensive gene replacement strategies (*Cruz and Beverley, 1990*), CRISPR/Cas9 has greatly improved loss-of-function experiments in *Leishmania* (*Beneke et al., 2017*; *Sollelis et al., 2015*; *Zhang et al., 2017*; *Zhang and Matlashewski, 2015*). Over the past 8 years, multiple CRISPR toolkits for kinetoplastids, including *Leishmania*, have emerged (*Bryant et al., 2019*; *Yagoubat et al., 2020a*; *Pal and Dam, 2022*), enabling specialised gene editing approaches, such as marker free (*Soares Medeiros et al., 2017*) or inducible gene deletions (*Damasceno et al., 2020*; *Rico et al., 2018*; *Yagoubat et al., 2020b*). In particular, the introduction of the PCR-based LeishGEdit toolbox (*Beneke et al., 2017*; *Beneke and Gluenz, 2019*; *Beneke and Gluenz, 2020*) has boosted the number of successful loss-of-function studies in *Leishmania* species (*Bryant et al., 2019*; *Yagoubat et al., 2020a*; *Jones et al., 2018*). LeishGEdit enabled bar-seq screens in-which hundreds of barcoded mutants are simultaneously phenotyped by pooling them together (*Beneke and Gluenz, 2020*; *Baker et al., 2021*; *Beneke et al., 2019*). This approach allows for precise deletion of single-copy or tandem arrayed multi-copy genes and since it produces complete knockouts, the number of false positives and negatives in any subsequent screening format is limited. Furthermore, the approach is scalable and it is therefore not surprising that this has led to a genome-wide loss-of-function screening proposal. Using the LeishGEdit method, the LeishGEM project (http://www.leishgem.org/) attempts to knockout nearly all genes in the *Leishmania mexicana* genome and plans to uncover key virulence factors by screening barcoded mutants across 170 pooled populations.

While these knockout pools will surely be a major resource for the research community in the future, their applications are limited. Cell pools are logistically difficult to share and it is unclear how well frozen pools recover and if they can be expanded after freezing without losing their uniformity. Also the composition of these pools cannot be changed and since each barcoded mutant needs to be created individually by replacing target genes with drug selectable marker cassettes (*Baker et al., 2021*; *Beneke et al., 2019*), bar-seq screens are most likely to be 'one-offs' on a genome-wide scale. Furthermore, aneuploidy in some *Leishmania* species can be a major challenge for gene replacement strategies as multiple rounds of transfection or isolation of clones may be required to target genes on multi-copy chromosomes. It is therefore unlikely that genome-wide bar-seq screens, such as the LeishGEM screen, could be repeated for *Leishmania* species that display extreme cases of aneuploidy, such as *L. donovani*. Using gene replacement approaches it is also not feasible to study multi-copy genes that are scattered over multiple chromosomes. These are major disadvantages of bar-seq screening. Hence, even though the LeishGEdit bar-seq approach and through LeishGEM available bar-seq pools, as well as novel RNAi tools provide a platform for possible genome-wide screens in *Leishmania* in the future, their applications remain extremely limited.

The lack of widely applicable high-throughput genome-wide screening platforms in *Leishmania* is in stark contrast to screening technologies that are available in mammalian cells (*Shalem et al., 2014*) and other parasites. For example, Cas9-expressing *Toxoplasma gondii* parasites can be transfected

with single-guide RNA (sgRNA) plasmid libraries to enable genome-wide CRISPR knockout screens (*Sidik et al., 2016*). One of the reasons why similar pooled CRISPR transfection screens in *Leishmania* species have not been performed is that *Leishmania* parasites, unlike mammalian cells or *T. gondii* parasites, lack the key components of the non-homologous DNA end joining (NHEJ) pathway, including DNA-PKcs, Ligase IV, and XRCC4 (*Passos-Silva et al., 2010*; *Zhang et al., 2022*). Following a Cas9-introduced double-strand break (DSB), DNA in mammalian cells and *T. gondii* parasites is mainly repaired by NHEJ (>70% in humans,>90% in *T. gondii* of all DSBs), allowing indels and frameshifts to be efficiently introduced in the target gene (*Fenoy et al., 2016*; *Fox et al., 2009*; *Gallagher and Haber, 2018*; *Hsu et al., 2014*; *Smolarz et al., 2014*). This greatly simplifies the knockout process, as just one ribonucleoprotein (RNP) complex, consisting of a Cas9 protein and an sgRNA, is required without the need for donor DNA templates. In contrast, *Leishmania* parasites rely on homologous recombination (HR), single-strand annealing, and microhomology-mediated end joining DNA repair pathways to repair Cas9-introduced DSBs. Without the addition of repair template DNA, this leads to large and unpredictable DNA deletions, increased cell death during DNA repair failures and prolonged repair times (*Zhang et al., 2017*; *Zhang and Matlashewski, 2015*; *Zhang et al., 2022*; *Zhang et al., 2019*). In consequence, CRISPR targeting in *Leishmania* is highly inefficient and the mutation rate is believed to be less than 1% without addition of donor DNA (*Zhang et al., 2017*; *Zhang and Matlashewski, 2015*; *Zhang et al., 2022*; *Zhang et al., 2019*). To overcome these limitations, there have been attempts to boost the CRISPR editing efficiency in *Leishmania* without requiring donor DNA. *Zhang et al., 2022* have recently attempted to reconstitute the NHEJ pathway of *Mycobacterium marinum* in *Leishmania*. While the authors claim that this improved DSB repair fidelity and efficiency in *Leishmania*, editing rates still remain low without the addition of donor DNA (less than 3% editing in selected non-clonal populations that express Cas9 and one sgRNA). These low editing rates mean in consequence that desired mutants can be only obtained through extra steps, which involve either the addition of donor DNA, labour-intensive isolation of clones or other means of selection, such as the selection of drug resistance-associated edits (*Beneke et al., 2017*; *Zhang et al., 2017*; *Zhang et al., 2022*). This makes pooled transfection loss-of-function screens currently unfeasible in *Leishmania* parasites.

Our aim was to overcome these limitations by bypassing the DSB repair machinery and instead directly editing suitable codons into STOP codons by using a CRISPR/Cas9 cytosine base editor (CBE) (*Figure 1A*). This strategy has previously been used for large-scale and genome-wide loss-of-function screens (*Kuscu et al., 2017*; *Després et al., 2020*). CBEs are comprised of a cytidine deaminase domain fused to an impaired form of Cas9. The Cas9 module provides precise targeting of the CBE to a site of interest, where the cytidine deaminase can directly convert cytosine to thymine without requiring DSBs, HR, or donor DNA templates (*Huang et al., 2021*). Specifically, the deaminase domain catalyses a deamination reaction, converting cytidine (C) to uridine (U). Uridine is then repaired to thymidine (T) during DNA replication. To prevent base excision repair, which would convert uridine back to cytidine, Cas9 nickases (nCas9), such as D10A (*Cong et al., 2013*), are additionally tethered to two monomers of a uracil glycosylase inhibitor (UGI) and further modifications of nuclear localisation signals (NLS), codon usage and the deaminase component have yielded highly efficient editors, such as the BE4max and AncBE4max (*Koblan et al., 2018*; *Komor et al., 2016*; *Komor et al., 2017*). In addition, CBE editing rates can be further improved through insertion of a non-sequence-specific single-stranded (ss) DNA-binding domain (DBD) from the Rad51 protein between the Cas9 nickase and the deaminase domain (namely hyBE4max) (*Figure 1A*; *Zhang et al., 2020*).

We successfully established the hyBE4max CBE in four *Leishmania* species and have created the online resource http://www.leishbaseedit.net/ to facilitate CBE primer design across 64 different kinetoplastid species. We show that our tool can be used to convert arginine, tryptophan, or glutamine codons into STOP codons with up to 100% editing efficiency in transfected cells that have been selected for CBE and sgRNA expression constructs. We demonstrate that single- and multi-copy genes can be depleted in *L. mexicana*, *L. major*, *L. donovani*, and *L. infantum* (the causative agents of cutaneous and visceral leishmaniasis) without requiring donor DNA and by expression of just one CBE-bound sgRNA. We then optimised the hyBE4max editor further by replacing the Rad51 ssDBD with a *L. major*-derived version, giving rise to hyBE4max-LmajDBD. Finally, we successfully use the hyBE4max and hyBE4max-LmajDBD CBE versions to perform a small-scale plasmid library delivered screen in *L. mexicana* and demonstrate that the viability phenotype of an essential gene can be

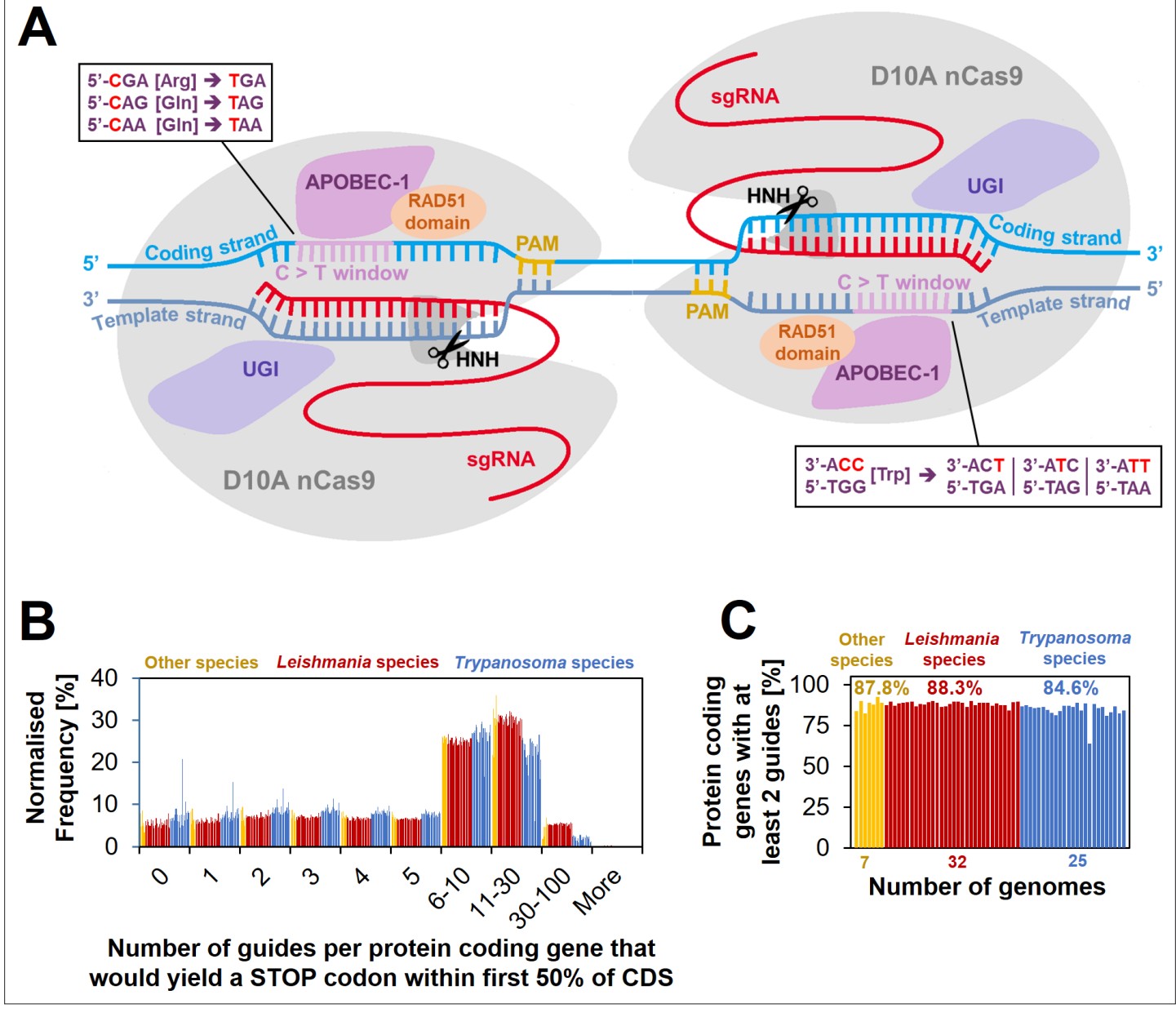

**Figure 1.** Cytosine base editing (CBE) in *Leishmania* and other kinetoplastids. (**A**) Schematic of hyBE4max CBE, consisting of D10A nCas9 with an HNH endonuclease domain (grey), a deaminase domain (APOBEC-1 [Apolipoprotein B mRNA editing enzyme catalytic polypeptide 1], pink), two uracil glycosylase inhibitor (UGI) monomers (purple), and a RAD51 ssDBD (orange). The editing window is located on the PAM sequence containing strand (yellow) at positions 4–12 (purple). Codons for arginine, tryptophan, or glutamine, can be edited into STOP codons, as shown in the text boxes. While one arginine and two glutamine codons are edited on the coding strand (light blue) into one STOP codon each, the one tryptophan codon on the template strand (dark blue) can be edited into three STOP codons. Suitable single-guide RNAs (sgRNAs) that would convert these four codons into STOP codons were designed for 64 kinetoplastid species. (**B**) The number of guides per protein-coding gene that yield a STOP codon within the first 50% of the CDS. (**C**) The percentage of protein-coding genes that could be targeted by at least two guides.

identified in a pooled format with our CBE method. To our knowledge, this is the first time a plasmid library transfection loss-of-function screen has been reported in *Leishmania*, enabling large-scale screens under various conditions in multiple *Leishmania* species in the future.

## Results
### Feasibility of cytosine base editing in kinetoplastids

To establish cytosine base editing in *Leishmania*, we decided to test the hyBE4max (*Zhang et al., 2020*) CBE version noted in the introduction. The insertion of the Rad51 ssDBD into a CBE not only increases the editing efficiency, but also extends the editing window, in-which cytidine to thymidine editing is efficient (without RAD51 domain the editing window ranges from position 4–7, with RAD51 domain from 4–12) (*Zhang et al., 2020*). This increases the number of possible C to T conversion targets and therefore the potential for genome-wide loss-of-function screening in *Leishmania* through the insertion of STOP codons. To test how many genes could be targeted on a genome-wide level, we first designed sgRNAs that would allow the introduction of STOP codons by C to T conversion within the 4–12 editing window. There is a total of four codons that can be converted into all three STOP codons (amber, ochre, and opal) by this approach (*Kuscu et al., 2017*). While one arginine and two glutamine codons are edited on the coding strand into one STOP codon each, there is one trypto-phan codon on the template strand that can be edited into three STOP codons (*Figure 1A*). We then filtered suitable sgRNAs that would introduce a STOP codon within the first 50% of the CDS, assuming that this would be sufficient to yield functional null mutants for most genes. We could on average design at least 2 sgRNAs per gene for 87.2% of all protein-coding genes in the genomes of 64 kineto-plastids (available through TritrypDB release 59 [*Aslett et al., 2010*]), including 32 *Leishmania* species (*Figure 1B, C*). For genes with more than two sgRNAs available, we generated a scoring matrix that ranked guides by their exact edit window (e.g. C to T conversion is more efficient at positions 4–8 than 8–12 [*Zhang et al., 2020*]) and based on how many STOP codons each guide would introduce. The primer design for all 64 genomes was then uploaded to our new primer design resource http://www.leishbaseedit.net/.

### Establishing cytosine base editing in *Leishmania*

Since this showed that CBE-mediated genome-wide loss-of-function screens in kinetoplastids would be feasible, we started to develop a strategy to express the hyBE4max CBE in *Leishmania*. First, we cloned the editor (*Zhang et al., 2020*) into pLdCH (*Zhang et al., 2017*), enabling episomal expression of the sgRNA and hyBE4max CBE via a ribosomal RNA (rRNA) promoter derived from *L. donovani* (*Figure 2A*). Then, we established a reporter assay to test the editor. A *L. major* clone was isolated, expressing a tdTomato (*Shaner et al., 2004*) reporter from the 18S rRNA locus (*Figure 2B, C*). In the following, two CBE sgRNAs were designed to induce a C to T conversion that results in a neutral substitution ('control' sgRNA; codon change but no amino acid change), while four other targeting guides were designed to yield a STOP codon through C to T conversion ('target' sgRNA). Three of these six guides were designed to target the coding strand, while the other three would target the template strand. Guide sequences were cloned into pLdCH-hyBE4max (*Figure 2A*) and then trans-fectants analysed 11, 21, and 33 days post transfection, while being under constant drug selection to maintain the episome. Strikingly, we observed strong knockdown effects of the tdTomato reporter across all four guides, designed to introduce a STOP codon ('Target 1' and 'Target 2' on template and coding strand in *Figure 2D*). For guide 1, targeting the template strand, 98.0% of transfected cells had no fluorescent signal as early as 11 days post transfection ('Target 1' on template strand in *Figure 2D*). In contrast, guides designed to only introduce neutral substitutions with no amino acid change ('Control' on template and coding strand in *Figure 2D*), did not alter the fluorescent signal of the reporter even 33 days post transfection. We also measured the doubling time of transfected cells. While the tdTomato reporter cell line showed an increase in doubling time compared to the wildtype cell line (from 5.6 to 7.5 h), additional transfection of the pLdCH-hyBE4max plasmid did not lead to a further increase in doubling time (*Figure 2E*), indicating no effect of the editor or guide expression on growth in *L. major* parasites.

To validate the observed changes of the tdTomato reporter, we then Sanger sequenced the reporter sequence before and after the editing (*Figure 3A*). Sanger sequencing reads were analysed by using

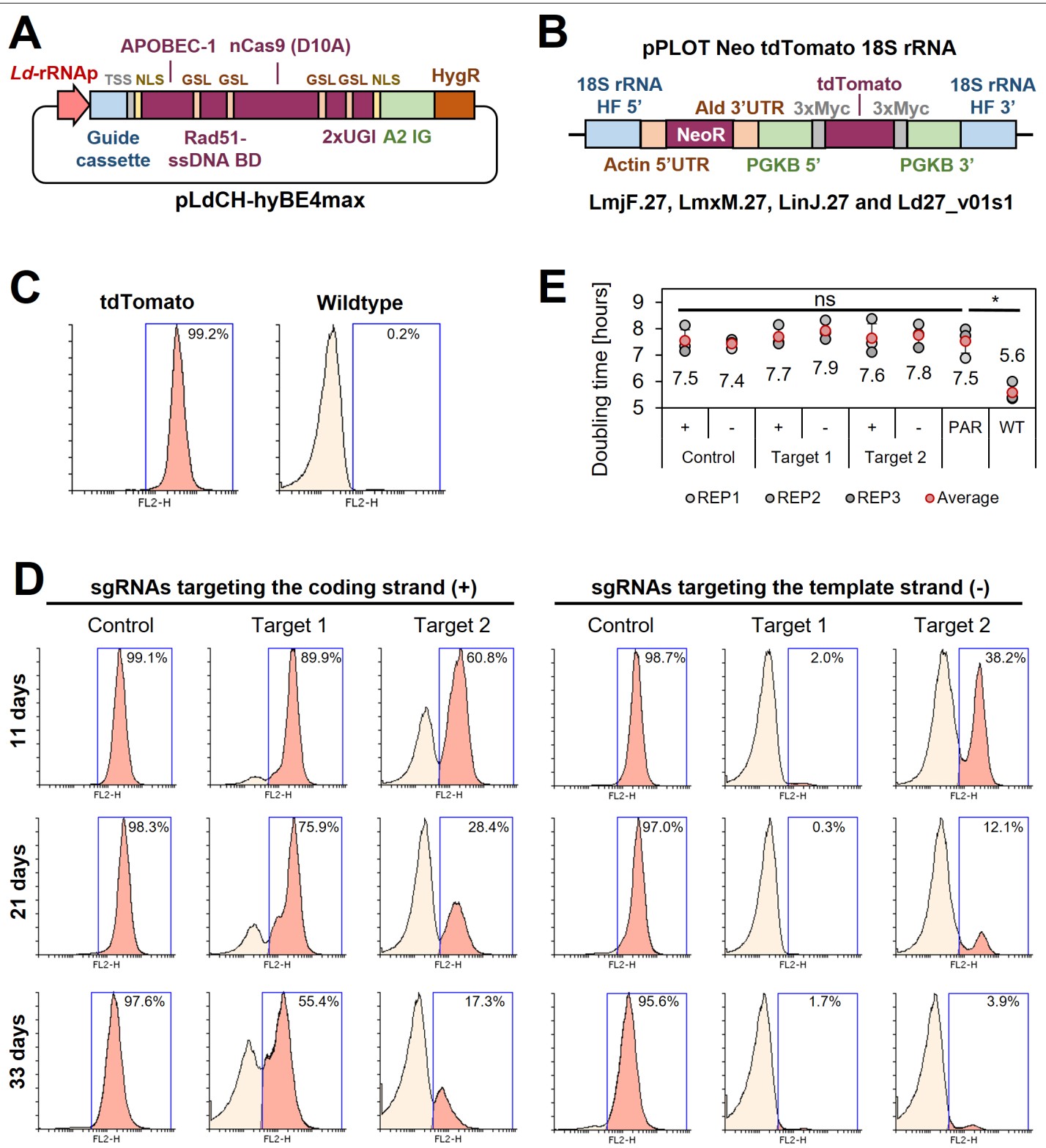

**Figure 2.** Cytosine base editing in *L. major* using an episomal hyBE4max expression constructs. (**A**) Schematic of pLdCH-hyBE4max plasmid, containing (from left to right) a *L. donovani*-derived ribosomal RNA (rRNA) promoter, single-guide RNA (sgRNA) expression cassette, hepatitis delta virus (HDV) ribozyme containing transsplice sequence (TSS), SV40 NLS, APOBEC-1 domain (ssDNA-specific cytidine deaminase), RAD51 ssDBD surrounded by two glycine–serine linkers (GSL), nCas9 (D10A) protein, two uracil glycosylase inhibitor (UGI) monomers tethered via GSL, nucleoplasmin NLS, *L. donovani*-derived A2 intergenic sequence, and hygromycin resistance marker. (**B**) Schematic of a pPLOT expression construct as described in **Beneke et al.,**

*Figure 2 continued on next page*

*Figure 2 continued*

*2017*, fused to two homology flanks for integration into the 18S rRNA locus on chromosome 27 (TriTrypDB [**Aslett et al., 2010**] release 59 coordinates: LmjF.27:989,898–991,525, LmjF.27:1,001,798–1,003,425; LmjF.27:1,013,829–1,015,456; LmjF.27:1,028,462–1,030,089; LmjF.27:1,041,477–1,043,104; LmjF.27:1,052,607–1,054,234; LmxM.27:982,640–984,268; LinJ.27:1,068,199–1,069,826; LinJ.27:1,078,759–1,080,386; LinJ.27:1,093,368–1,094,995; Ld27_v01s1:1,021,892–1,020,265). (**C**) FACS (fluorescence-activated cell sorting) plot showing tdTomato-expressing and wildtype *L. major* parasites. (**D**) FACS plot of tdTomato-expressing *L. major* parasites, transfected with pLdCH-hyBE4max-tdTomato targeting plasmids (see description main text). Cells were analysed 11, 21, and 33 days post transfection. Percentages in (**C**) and (**D**) represent the remaining proportion of tdTomato-expressing cells. (**E**) Doubling times for analysed parasites in (**C**) and (**D**). Error bars show standard deviations of triplicates. PAR: tdTomato-expressing cells; WT: wildtype cells. Asterisks indicate Student's *t*-test: *p > 0.05.

The online version of this article includes the following source data and figure supplement(s) for figure 2:

**Figure supplement 1.** Confirming identity of *Leishmania* species used in this study.

**Figure supplement 1—source data 1.** Data used for table shown in *Figure 2—figure supplement 1*.

**Figure supplement 2.** Cytosine base editing in *L. donovani*, *L. mexicana*, *L. infantum*, and *L. major* using episomal hyBE4max expression constructs.

**Figure supplement 3.** Alignment of *Leishmania* ribosomal RNA (rRNA) promoter.

ICE (inference of CRISPR edits; Sythego), allowing to quantify the discordance (editing rate) between the normalised edited and non-edited read for the entire sequenced region (from 2.7% to 25% of the tdTomato CDS) (**Figure 3B**). For all six guides, editing was only observed within the guide target sequence and not elsewhere in sequenced regions (**Figure 3A, B**). While selected C to T conversions within the editing window (positions 4–12) were up to 100% effective (100% editing rate) for three of the six guide target sequences, not all possible C to T conversion targets were effectively edited. For example, three C to T conversions were 100% effective for 'Target 1' on the template strand at positions 6–8, yielding a STOP codon at the intended position. However, the fourth conversion target of that guide at position 12 was not edited (**Figure 3A**), confirming previously observed differences in editing efficiencies across the editing window (efficiency higher at positions 4–8 than 8–12 [**Zhang et al., 2020**]). Strikingly, both control guides, which did not show any changes in the reporter expression (**Figure 2D**) as they were designed to only change the DNA sequence but not the amino acid sequence, showed also high editing rates (100% for Ctrl(+), 79.2% and 45.9% for Ctr(−)), confirming that the hyBE4max CBE was highly specific even though expressed for 33 days (**Figure 3A, B**).

Since editing rates did not reach 100% for all guides after 33 days, we thought that stable expression of the hyBE4max CBE would increase editing rates and inserted hyBE4max into the β-tubulin locus using the pTB007 plasmid as a vehicle (**Beneke et al., 2017**; **Figure 3—figure supplement 1A**). This also allowed to express sgRNAs under the control of the T7 RNAP promoter either via transcription from short DNA templates or from epsiomes (**Figure 3—figure supplement 1B**). While the latter strategy increased editing during our tdTomato reporter assay, with all six guides reaching nearly 100% editing rate (**Figure 3—figure supplement 1C–E**) and controls showing no effect on the reporter expression (**Figure 3—figure supplement 1D**), doubling times significantly increased once episomal-expressed sgRNAs were transfected into the cell line expressing constitutively hyBE4max and T7 RNAP (**Figure 3—figure supplement 1F**). We also tested another CBE version (namely AncBE4max [**Koblan et al., 2018**]) that, unlike hyBE4max (**Zhang et al., 2020**), does not have the Rad51 ssDBD (**Figure 3—figure supplement 2A**). While we observed also high editing rates of 100% for nearly all sgRNAs when targeting tdTomato (**Figure 3—figure supplement 2B–E**), we saw strong editing outside of the guide targeting window (**Figure 3—figure supplement 2F**), depleting the tdTomato reporter signal even in control cells, where only neutral substitutions were intended with no amino acid change (**Figure 3—figure supplement 2D**). Meanwhile, editing outside the guide target sequence was not observed when using the hyBE4max CBE (**Figure 3—figure supplement 2F**).

We concluded from these initial results that episomal expression of the sgRNA and hyBE4max CBE is a suitable approach for mutant generation. While stable expression systems did not affect growth before expressing the sgRNA and may be attractive too after further optimisation, an episomal expression system seemed to be much more flexible, allowing to transfect wildtype parasites without further modifications. To test the flexibility of our system, we therefore next performed our reporter assay in *L. donovani*, *L. mexicana*, and *L. infantum*, and repeated it in *L. major*. First, we confirmed the identity of each *Leishmania* species by sequencing the LPG2 locus (**Figure 2—figure supplement 1A, B**; **Akhoundi et al., 2017**) and generated tdTomato-expressing clones (**Figure 2B**). We then transfected all four species with pLdCH-hyBE4max plasmids, targeting the tdTomato reporter as in

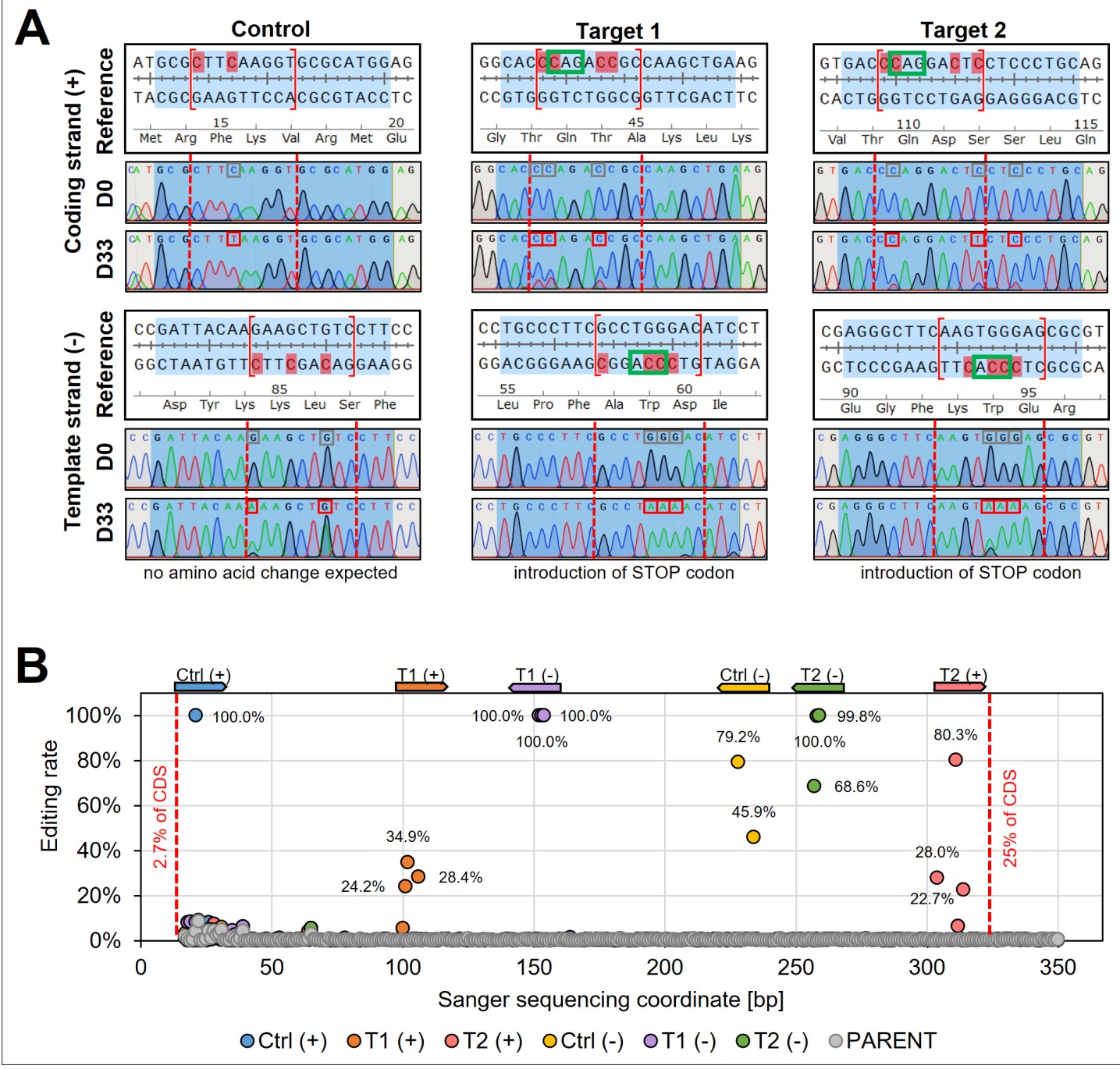

**Figure 3.** Sanger sequencing following cytosine base editing in *L. major*. (**A**) Alignment of reference sequences (Reference) with Sanger sequencing trace plots before transfection of pLdCH-hyBE4max-tdTomato targeting plasmids (D0) and 33 days after transfection (D33). Six guide loci are shown as described in the main text. Blue shading: 20nt guide target sequence. Red dotted lines: 4–12 editing window. Red marked nucleotides: expected cytosine to thymine conversion. Red boxed nucleotides: edited nucleotides. (**B**) Discordance between D0 and D33 trace plots from (**A**) was measured and plotted as editing rate versus Sanger sequencing coordinates. Each data point represents one C to T conversion. The position and orientation of each guide are indicated above, with the first guide sequence starting after 2.7% of the CDS and the last guide sequence ending after 25% of the CDS.

The online version of this article includes the following figure supplement(s) for figure 3:

**Figure supplement 1.** Cytosine base editing in *L. major* using stable hyBE4max and episomal single-guide RNA (sgRNA) expression.

**Figure supplement 2.** Cytosine base editing in *L. major* using stable AncBE4max and episomal single-guide RNA (sgRNA) expression.

experiments above, and measured the reporter signal 16 days post transfection. First, we assessed whether we could reproduce results in *L. major*. For all four repeated guide transfections ('Control (−)', 'Target 1 (+)', 'Target 1 (−)', and 'Target 2 (−)') similar results were obtained (*Figure 2D* and *Figure 2—figure supplement 2A*) with again almost 100% elimination of the tdTomato reporter signal in cells transfected with guide 1 targeting the template strand ('Target 1 (−)' in *Figure 2—figure supplement 2A*). While we also achieved full depletion of the tdTomato reporter for selected guides in the other three species, the reporter was strongest depleted in *L. donovani*, followed by *L. mexicana*, *L. infantum*, and *L. major* (*Figure 2—figure supplement 2A*). It is possible that this result was caused by a potentially different number of integrated tdTomato expression constructs, as we did not assess the exact tdTomato reporter copy number in each species. However, given that the pLdCH-hyBE4max expression cassette is driven by an *L. donovani* rRNA promoter, which is diverse compared to other *Leishmania* rRNA promoters (*Martínez-Calvillo et al., 2001*; *Yan et al., 1999*; *Figure 2—figure supplement 3*), a species-dependent result is anticipated. A pattern was also observed when assessing the doubling time of all four transfected species. Expression of the CBE and sgRNA did not affect *L. major* and *L. mexicana* growth but it increased significantly doubling times in *L. donovani* and *L. infantum*, suggesting that high CBE and sgRNA expression levels may be toxic for these parasites (*Figure 2—figure supplement 2B*). Conversely, this was not reflected when assessing the expression levels of hyBE4max in all four species. While protein degradation made an exact quantification of hyBE4max difficult, western blots indicated that the highest expression of hyBE4mas was seen in *L. major*, followed by *L. infantum*, *L. mexicana*, and *L. donovani* (*Figure 2—figure supplement 2C*).

## CBE targeting of single- and multi-copy genes in *Leishmania*

Despite the slight toxicity observed in *L. donovani* and *L. infantum*, we concluded that our system was efficiently and specifically converting C to T within the target window, thereby introducing STOP codons in all four *Leishmania* species. This prompted us to generate proof of principle mutants in wild-type cells next. To test our CBE system we decided to target the single-copy genes PF16 (encoding a central pair protein of the axoneme and essential for flagellar motility [*Beneke et al., 2017*; *Martel et al., 2017*]), IFT88 (encoding a subunit of the IFT-B complex and essential for flagellar assembly [*Beneke et al., 2019*]), and the miltefosine transporter (MFT, encoding a P-type ATPase that is linked to miltefosine resistance in *Leishmania* [*Seifert et al., 2007*]), as well as the multi-copy gene PFR2 (encoding a key component of the paraflagellar rod and required for normal flagellar motility [*Beneke et al., 2019*; *Maga et al., 1999*]). All four genes were targeted with four guides each, which were cloned into pLdCH-hyBE4max plasmids (*Figure 2A*). While we designed PF16 and IFT88 guides to be universal for multiple *Leishmania* species, guides for MFT and PFR2 were species specific for *L. donovani* and *L. mexicana*, respectively. In addition, guides targeting PFR2 were designed to be universal for all three PFR2 copies (A, B, and C). We then individually transfected plasmids targeting PF16 into *L. donovani*, *L. mexicana*, *L. infantum*, and *L. major*, while plasmids targeting IFT88 were only transfected into *L. infantum* and *L. major*. Meanwhile, plasmids targeting MFT were only transfected into *L. donovani* and plasmids targeting PFR2 only into *L. mexicana*.

Following transfection, each cell line was subjected to motility analysis and the velocity of tracked cells was determined (*Wheeler, 2017*; *Figure 4* and *Supplementary file 1*). We measured a significant decrease in motility for three (PF16-1, PF16-2, and PF16-3) of the four guides targeting PF16 in *L. donovani* at 14 days post transfection. In *L. mexicana* and *L. infantum* only one guide (PF16-3) yielded a measurable decrease in swimming speed, while in *L. major* no significant effect could be determined. For guides targeting PFR2, MFT, and IFT88, no significant decrease in swimming speed was detected in *L. mexicana*, *L. donovani*, and *L. major*, respectively (*Figure 4A* and *Supplementary file 1*). However, out of the four IFT88 targeting guides, one (IFT88-1) reduced the swimming speed in the transfected *L. infantum* population (*Figure 4A* and *Supplementary file 1*) and also yielded the expected flagellar assembly phenotype (*Figure 4—figure supplement 3*).

We then cultured the cells for another 14 days, keeping them under constant drug selection to maintain the guide carrying pLdCH-hyBE4max episome. As expected, we saw for most cell lines a further increase in proportion of cells that showed a decrease in swimming speed when PF16 was targeted. For *L. donovani* three of the four PF16 targeting guides (PF16-1, PF16-2, and PF16-3) still yielded significantly decreased motility, while for MFT no significant effect was observed (*Figure 4B*, *Figure 4—figure supplement 1A*, and *Supplementary file 1*). This confirmed that observed motility

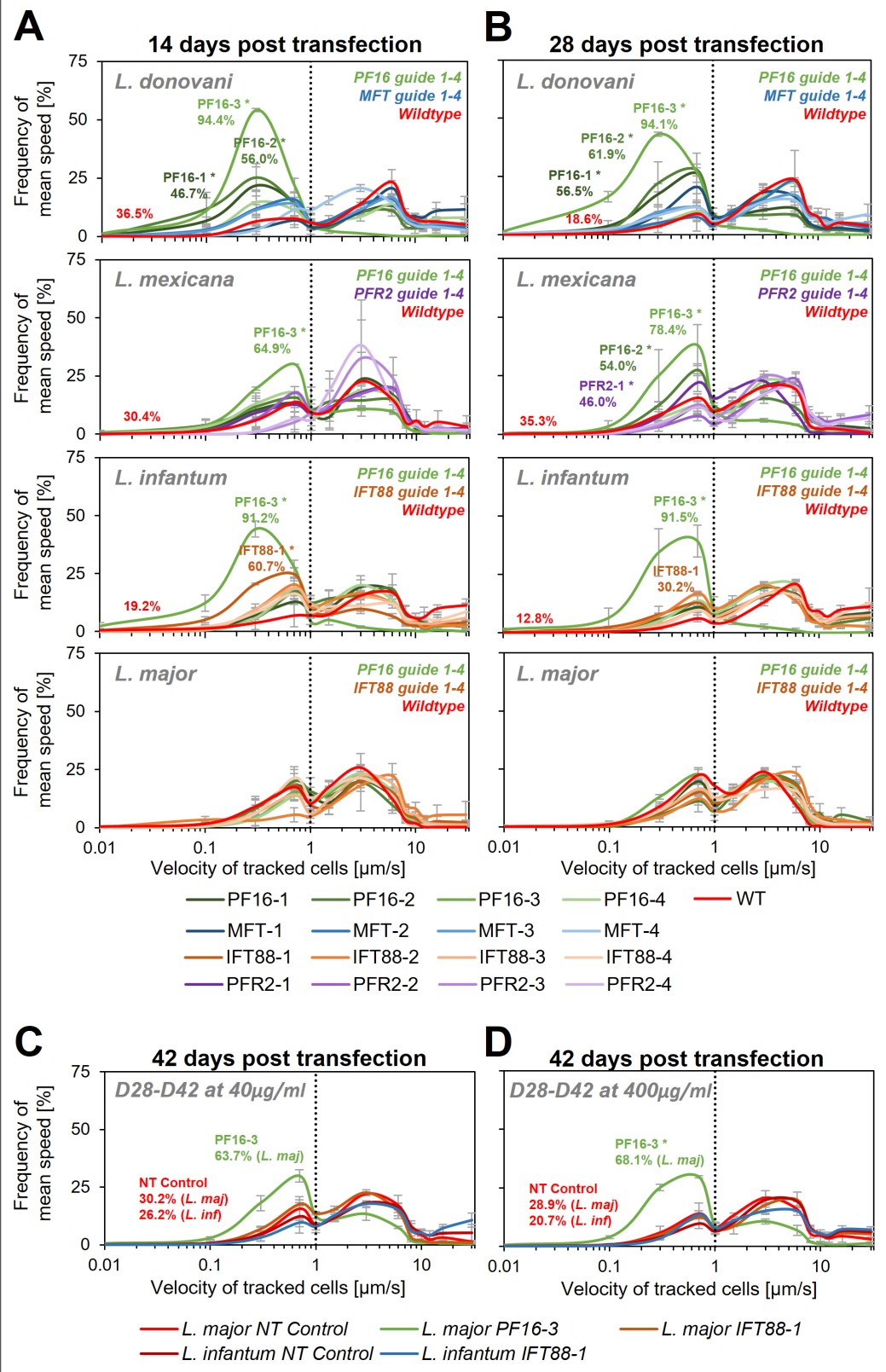

**Figure 4.** Motility analysis following hyBE4max-mediated cytosine base editing. *L. donovani, L. mexicana, L. infantum*, and *L. major* wildtype parasites were transfected with pLdCH-hyBE4max-sgRNA expression plasmids, targeting PF16, MFT, PFR2, and IFT88 with four guides each (see main text description). Selected non-clonal populations were subjected to motility analysis and the frequency of the velocity of tracked cells was plotted

*Figure 4 continued on next page*

*Figure 4 continued*

(**A**) 14 days, (**B**) 28 days, and (**C, D**) 42 days post transfection. Cells were selected with 40 µg/ml hygromycin B until 28 days post transfection and then with both, (**C**) 40 and (**D**) 400 µg/ml. Each population was analysed using a Cramér-von Mises Test to detect any shift in the population distribution towards lower speed. Highlighted curves are marked with an asterisks when that shift was significant (for *L. donovani*, *L. mexicana*, and *L. infantum*: *p > 0.05, for *L. major*: *p > 0.01). For populations with a significant shift and for wildtype and NT controls, the overall percentage of tracked cells that have a velocity of less than 1 µm/s is highlighted. NT: non-targeting control, a wildtype transfected parasite, harbouring a pLdCH-hyBE4max-tdTomato-targeting-guide (Target 1 (−)) plasmid. Mean speed was measured in duplicates 14 days post transfection and in triplicates 28 and 42 days post transfection. Error bars show standard deviation between these replicates.

The online version of this article includes the following figure supplement(s) for figure 4:

**Figure supplement 1.** Motility tracks following hyBE4max-mediated cytosine base editing.

**Figure supplement 2.** Motility tracks following hyBE4max-mediated cytosine base editing under different selection dosis.

**Figure supplement 3.** Flagellum assembly phenotype following hyBE4max IFT88 targeting.

**Figure supplement 4.** Increased drug selection concentration elevates the protein expression level of hyBE4max in *L. major* but not *L. infantum*.

phenotypes were specific to the targeted gene. In *L. mexicana* two of the four PF16 targeting guides (PF16-2 and PF16-3) and one of the PFR2 targeting guides (PFR2-1) showed now a significant effect (*Figure 4B*, *Figure 4—figure supplement 1A*, and *Supplementary file 1*). In *L. infantum*, the percentage of cells showing decreased mutant speed was now lower for the IFT88 targeting guide (IFT88-1), which is expected as the deletion of IFT88 in *L. mexicana* has been shown to reduce the doubling time (*Beneke et al., 2019*). *L. infantum* parasites with a deleterious IFT88 mutation are therefore most likely outcompeted from the mutant population (*Figure 4B*, *Figure 4—figure supplement 1A*, and *Supplementary file 1*).

Since we still did not observe a measureable effect on any of the transfected *L. major* parasites at 28 days post transfection, we cultured parasites for another 14 days. We increased the hygromycin B concentration by 10-fold, in an attempt to force the cells to ramp-up their episomal expression of pLdCH-hyBE4max. We also continued the incubation of *L. infantum* parasites, transfected with a guide targeting IFT88 (IFT88-1). As expected (*Beneke et al., 2019*), the proportion of cells showing a motility phenotype in the IFT88 targeted *L. infantum* population decreased further (*Figure 4C*, *Figure 4—figure supplement 2B*, and *Supplementary file 1*). For *L. major* parasites, we finally could detect cells with a motility phenotype within the PF16 guide transfected population (PF16-3). However, the transfected IFT88 guide (IFT88-1) still did not induce any measurable effect in *L. major* (*Figure 4C*, *Figure 4—figure supplement 2B*, and *Supplementary file 1*). Meanwhile, increased drug selection did not significantly increase the proportion of cells affected (*Figure 4D*, *Figure 4—figure supplement 2B*, and *Supplementary file 1*), even though we could detect an increase in expression of the hyBE4max CBE protein for *L. major* (*Figure 4—figure supplement 4*).

To verify that observed phenotypes were induced by C to T editing, we then Sanger sequenced the guide targeting loci of PF16, IFT88, and PFR2 in respective targeted *Leishmania* species (from time point 28 days post transfection) (*Figure 5* and *Supplementary file 2*). Observed motility phenotypes correlated well with editing rates. For example, guide PF16-3, yielding the strongest decrease in motility across all species, showed high editing rates in all four species with up to 100% C to T conversion. Guide PF16-2, which only showed a motility phenotype in *L. mexicana* and *L. donovani*, also only showed editing in those two species and guide PF16-1, which only showed a decrease in mutant speed in *L. donovani*, also just showed editing in *L. donovani* (*Figures 4B and 5A*). Surprisingly, we saw high editing rates for three of the four IFT88 targeting guides in *L. infantum* and *L. major* but only guide IFT88-1 resulted in a measurable motility and flagellar assembly phenotype in *L. infantum* (*Figures 4B and 5B* and *Figure 4—figure supplement 3*). Analysing the Sanger sequencing trace plots further, we noticed that only guide IFT88-1 induced a mutation that resulted in a STOP codon, while non-STOP codon mutations became dominant for the other two guides (*Supplementary file 2*), presumably due to the known IFT88 growth phenotype (*Beneke et al., 2019*). For guides targeting PFR2 in *L. mexicana*, only guide PFR2-1 was efficient, which was also the only guide resulting in decreased motility (*Figures 4B and 5C*). To confirm that indeed, the knockdown of PFR2 was successful, we analysed the

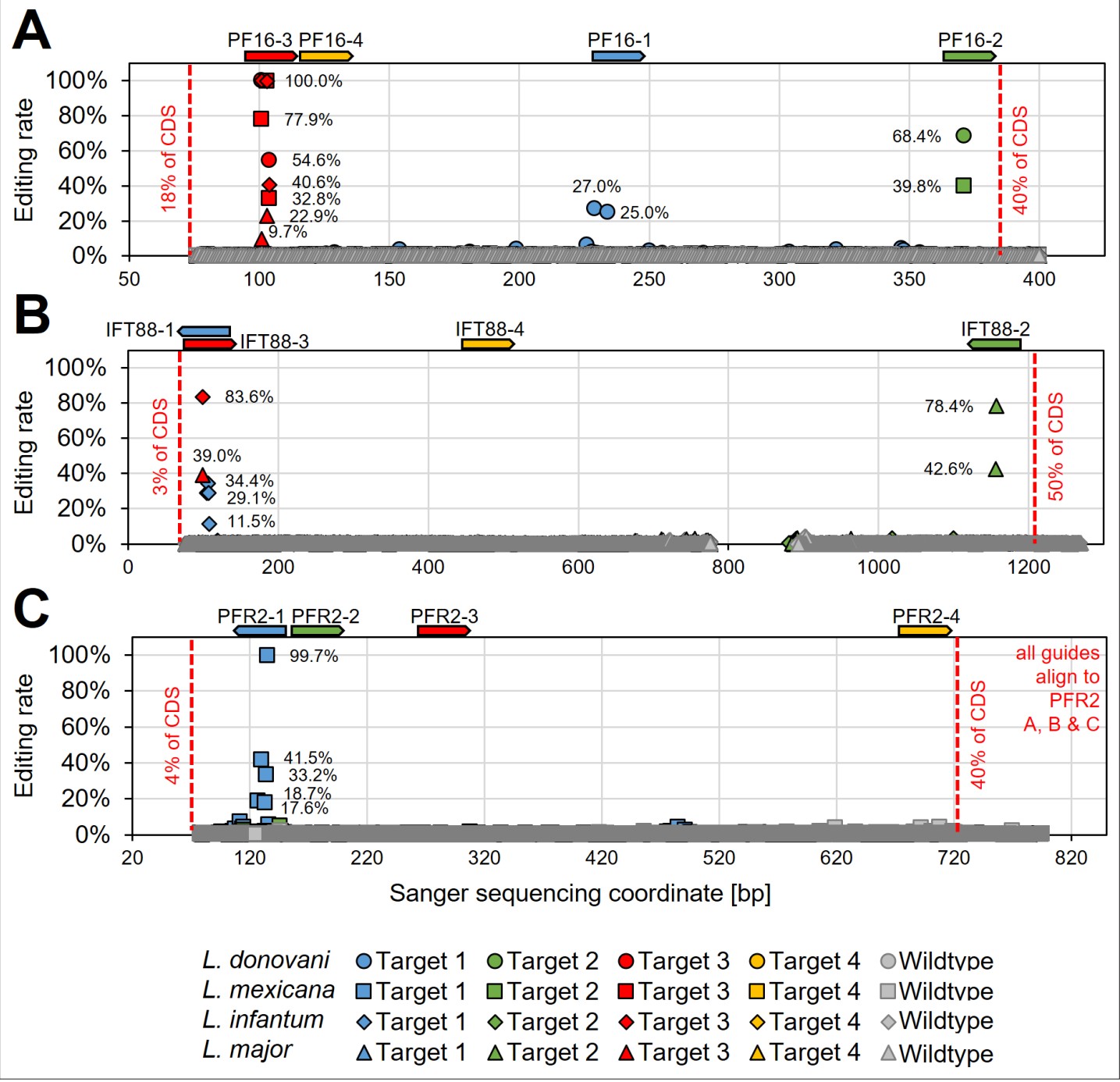

**Figure 5.** Determining hyBE4max editing rates following targeting of single- and multi-copy genes. *L. donovani*, *L. infantum*, *L. mexicana*, and *L. major* wildtype parasites were transfected with pLdCH-hyBE4max-sgRNA expression plasmids, targeting (**A**) PF16, (**B**) IFT88, and (**C**) PFR2 with four guides each (see main text description). Discordance from Sanger sequencing traces before transfection (wildtype) and 28 days after transfection (Targets 1, 2, 3, and 4) was measured and plotted as described in *Figure 3B*. The position and orientation of each guide are indicated above.

The online version of this article includes the following figure supplement(s) for figure 5:

**Figure supplement 1.** Effective PFR2 knockdown using hyBE4max for cytosine base editing.

expression of the targeted protein by western blot, confirming a strong knockdown only for guide PFR2-1 (~97% knockdown compared to wildtype) (*Figure 5—figure supplement 1*). It is important to highlight that this strong knockdown was achieved in non-clonal populations that had been selected to express the hyBE4max and only one sgRNA. There was no requirement for additional donor DNA or isolation of clones. This is despite the fact that PFR2 is a three-copy gene in a tandem array.

Since guides targeting MFT in *L. donovani* did not result in decreased swimming speed (as expected), we wanted next to assess whether they would instead confer resistance to miltefosine, an orally given antileishmanial drug. To determine the suitability of our method for drug screening applications, such as positive drug resistance screens, we also wanted to test whether miltefosine-resistant parasites could be enriched in the mutant population by treating them with a low dose of miltefosine prior to assessing their actual resistance to the drug itself. To test this, wildtype parasites and parasites expressing MFT targeting guides were treated for 48 hr with 20 µM miltefosine (pre-treated parasites) from 16 days post transfection onwards. Then cells were washed and cultured for another 72 hr. In the following, pre-treated and non-pre-treated parasites were incubated with different doses of miltefosine for 48 hr and subsequently cell viability was measured by using an MTT assay (*Figure 6A* and *Supplementary file 3*). Pre-treated with miltefosine or not, wildtype cells were only resistant to small doses of miltefosine (5 and 20 µM). In comparison, parasites transfected with pLdCH-hyBE4max plasmids that allowed to express MFT targeting guides became significantly resistant to high doses of miltefosine (50 and 80 µM) (guide MFT-1 and MFT-3 significantly different in *Figure 6C*). As expected, when pre-treated with miltefosine the proportion of viable parasites at high miltefosine concentrations then rose further and three out of four guides were able to confer resistance (guide MFT-1, MFT-2, and MFT-3 significantly different in *Figure 6B*). In addition, viability rates correlated well with high editing rates for all these three guides. For example, guide MFT-3, which conferred the highest level of resistance to miltefosine (*Figure 6B, C*), resulted also in the highest editing rate of all MFT targeting guides (up to 81.9% for non-pre-treated cells and 86.5% for pre-treated cells) (*Figure 6D, E*). Further analysis of trace plots also highlighted that the introduced mutations resulted in the intended premature STOP codon (*Supplementary file 2*). This demonstrates that our method can potentially be used for positive and antileishmanial drug-resistant screening.

Finally, we also analysed trace plots along the entire Sanger sequencing range to verify the editing window. Out of all 26 guides used in this study, including tdTomato targeting guides from above, 24 guides resulted in editing exclusively within the expected hyBE4max 4–12 editing window (*Zhang et al., 2020*). One guide showed editing activity within 3nt range of the 4–12 editing window (tdTomato guide Target 2 (+); *Figure 3A*) and one guide resulted in editing within 6nt range of the guide sequence (PF16-1 in *L. donovani*; *Supplementary file 2*). However, editing rates of these two guides were relatively low (23% and 25–27%, respectively), while editing within the 4–12 editing window reached up to 100%.

## A *Leishmania*-optimised CBE for targeting essential genes in pooled transfection screens

While these proof of principle mutants demonstrate that CBEs can be used to efficiently introduce STOP codons in *Leishmania* genes, allowing to generate functional null mutant populations by using just one sgRNA, we wanted to further improve our method by modifying the Rad51 ssDBD. Comparing the Rad51 ssDBD from *Leishmania* with the human version, the *Leishmania* version contains an N-terminal extension that is bound to the conserved Rad51 ssDBD part through a glutamine-rich repeat (*Figure 7—figure supplement 1A*). Glutamine-rich repeats, known to be enriched in transcription factors, can increase DNA–protein binding by forming hydrogen bonds via their amide site chains (*Perutz et al., 1994*). Since *L. major* had the longest glutamine-rich repeat among our tested *Leishmania* species, we decided to replace the Rad51 ssDBD in pLdCH-hyBE4max with an *L. major*-derived Rad51 ssDBD, giving rise to pLdCH-hyBE4max-LmajDBD (*Figure 7—figure supplement 1B*). We then chose *L. mexicana* to test our *Leishmania*-optimised CBE as in experiments above editing rates in *L. mexicana* were higher than in *L. major* while not affecting parasite growth. First, we repeated our tdTomato reporter assay and saw an increase in the knockdown effect for the *Leishmania*-optimised CBE version for two of the four targeting guides at 16 days post transfection (1.6-fold increase for guide Target 2 on the coding strand and 5.3-fold increase for guide Target 2 on the template strand).

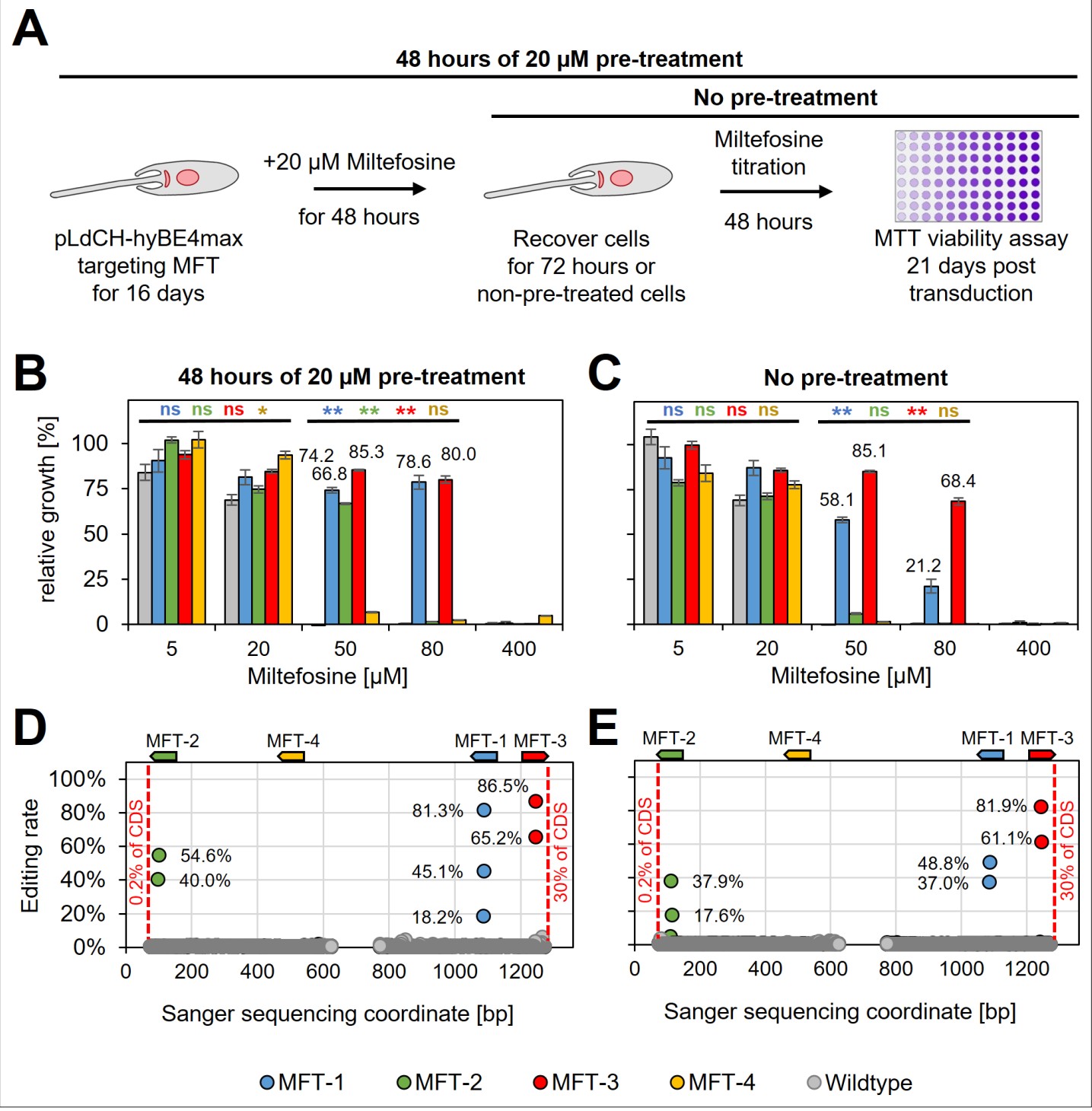

**Figure 6.** HyBE4max targeting of the *L.donovani* miltefosine transporter (MFT). (**A**) *L. donovani* wildtype parasites were transfected with pLdCH-hyBE4max-sgRNA expression plasmids, targeting MFT with four guides each. Sixteeen days post transfection, transfected and non-transfected parasites were treated for 48 hr with 20 µM miltefosine, then washed and cultured for another 72 hr (pre-treated parasites). (**B**) Pre-treated and (**C**) non-pre-treated parasites were subjected to different doses of miltefosine (5, 20, 50, 80, and 400 µM) and relative growth between 'no drug controls' and each different dose was measured using an MTT cell viability assay. Asterisks indicate one-way analysis of variance (ANOVA) test with post hoc Tukey honestly significant difference (HSD), comparing each mutant and wildtype: *p > 0.05, **p > 0.01 (blue: MFT-1; green: MFT-2; red: MFT-3; yellow: MFT-4). Error bars show standard deviations of triplicates. Discordance from Sanger sequencing traces before transfection (wildtype) and 21 days after transfection (MFT-1, 2, 3, and 4), from either (**D**) pre-treated and (**E**) non-pre-treated populations, was measured and plotted as described in *Figure 3B*. The position and orientation of each guide are indicated above.

For all other guides, including control guides that were designed not to yield a STOP codon, no major difference between both CBE versions could be detected (*Figure 7—figure supplement 1C*).

Encouraged by the slightly improved editing rate, we then decided to test whether our method could also be used to target essential genes. Our earlier results indicated that the individual targeting of IFT88, a gene required for normal proliferation (*Beneke et al., 2019*), results in outcompeting of the introduced STOP codon mutation over time (*Figure 4*). This outcompeting process should in-turn result in reduced growth of the overall mutant population. Therefore, our assumption was that the viability phenotype of an essential gene could especially be detected in a pooled screening format. To test this, we pooled all annealed guide oligos from experiments above, targeting IFT88, PF16, MFT, PFR2, and tdTomato, and generated an oligo pool (*Figure 7A*). In addition, we added four guides to this oligo pool that target the Cdc2-related kinase 3 (CRK3), an essential kinase that has been used as a proof of principle mutant for essential genes in previous studies (*Yagoubat et al., 2020b*; *Duncan et al., 2016*). The prediction was that CRK3 guides would be strongly depleted from the pool over time. In addition, we wanted to use this experiment as an opportunity to compare again our *Leishmania*-optimised CBE (pLdCH-hyBE4max-LmajDBD) with the non-optimised version (pLdCH-hyBE4max). We therefore cloned the oligo pool, consisting of these 26 guides, into both plasmids (*Figure 7A*). We then transfected this library in triplicates into *L. mexicana* wildtype parasites, so that the four guides targeting MFT, originally designed for *L. donovani*, and the six guides targeting tdTomato, having no target in wildtype cells, would serve as non-targeting controls. Meanwhile, IFT88, PF16, PFR2, and CRK3 would be targeted by four guides each.

Immediately after library transfection, we also assessed the representation of each guide within the library by subjecting an aliquot of each library and replicate to serial dilutions. This showed an average representation of ~15 transfectants/guide (*Figure 7—figure supplement 2A*), which is three times as high as typically used in pooled RNAi libraries in *T. brucei* (*Morris et al., 2002*). Selected clones from these serial dilutions were then also Sanger sequenced to determine whether parasites would harbour one or multiple sgRNAs. This showed that most transfectants had just been transfected with a single episome. However, three out of eight sequenced clones maintained two different plasmids and by that expressed two sgRNAs (*Figure 7—figure supplement 2B*). This meant that for most transfectants only the effect of one guide per transfectant is measured. However, there must have been also transfectants in the pool with simultaneously induced mutations on multiple genes. Since we could not find clones that harboured more than two different guide sequences and since these seemed to be randomly mixed, we concluded that effects of multi-gene targeting in one transfectant should be minimal within the mutant pool.

Hence, we proceeded with the viability screen and analysed by Illumina sequencing sgRNA abundances from transfected libraries at 12 and 21 days post transfection (*Figure 7A*). To compare the sgRNA abundance across the screen, we assumed that each plasmid had an equal chance of being transfected and therefore *z*-scored the ratio between normalised sgRNA reads at plasmid level (before transfection) and days 12 and 21. Strikingly, we found two of the four CRK3 targeting guides (CRK3-3 and CRK3-4) to be significantly depleted at days 12 and 21 in the *Leishmania*-optimised CBE library (pLdCH-hyBE4max-LmajDBD), while only one CRK3 guide (CRK3-3) was significant depleted from the non-optimised CBE library (pLdCH-hyBE4max). All other guides, including non-targeting guides, did not significantly change in their representation within the pool over time (*Figure 7B* and *Supplementary file 4*). This demonstrates that our method can not only be used to target essential genes but also to pursue loss-of-function screens in *Leishmania* via delivery of plasmid libraries.

## Discussion

### Advantages and disadvantages of cytosine base editing in *Leishmania*

Here, we report our new LeishBASEedit CBE toolbox, allowing to knockdown single- and multi-copy genes in *Leishmania* populations by expressing hyBE4max and just one sgRNA. Our method to convert selected codons into STOP codons by using a base editor bypasses the requirement of DSBs, HR, donor DNA, or isolation of clones in order to obtain functional null mutants. Since our technique resembles a gene silencing approach, it is applicable for targeting essential genes in pooled transfections. Most importantly, this enables genetic screening in wildtype parasites (no engineered cell lines required) through a pooled plasmid library transfection format. While most of these points also

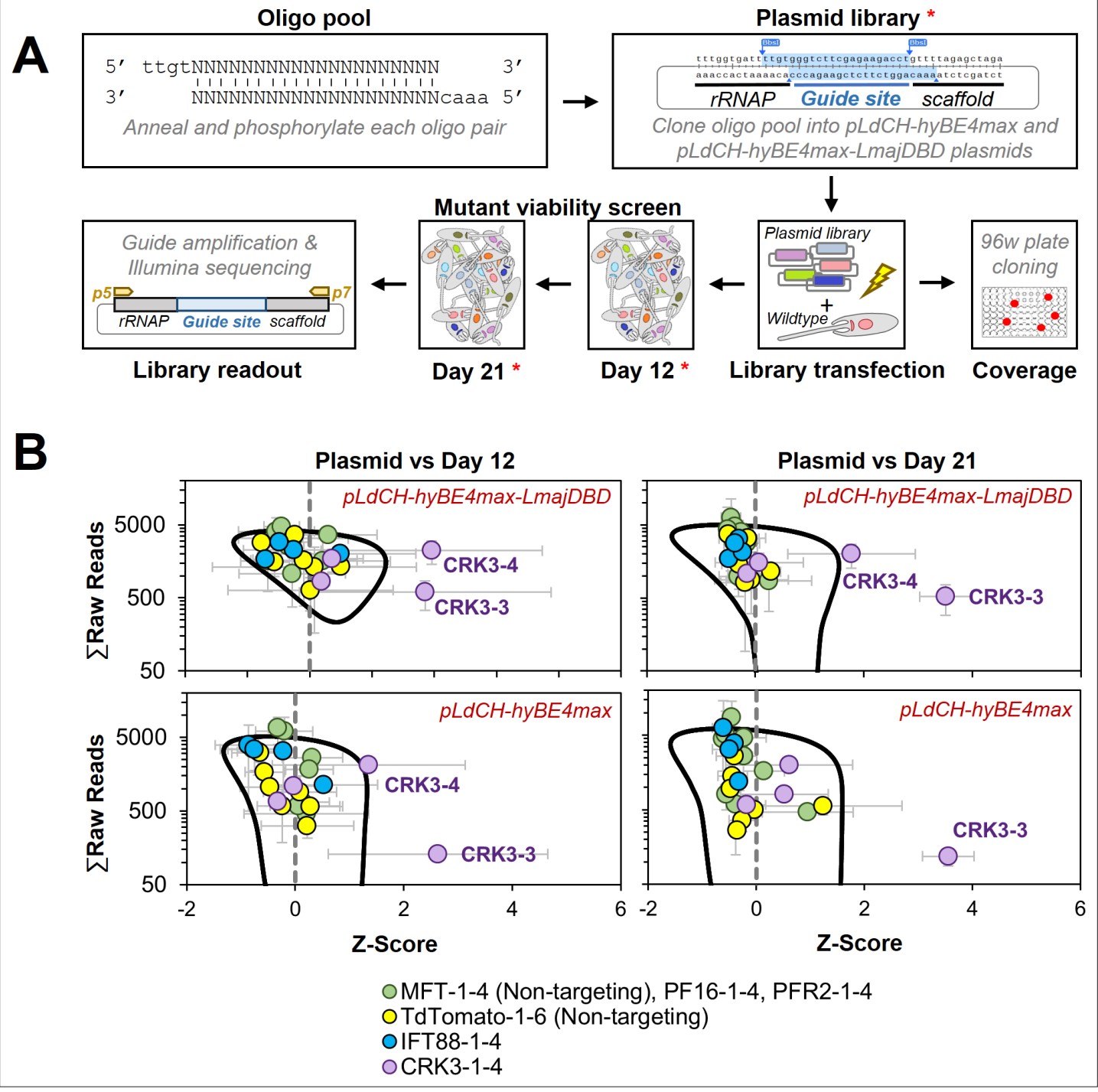

**Figure 7.** Targeting essential genes in a plasmid library delivered loss-of-function screen. (**A**) Schematic of hyBE4max loss-of-function screen in *Leishmania*. Oligo pairs were annealed and phosphorylated, then pooled and cloned into pLdCH-hyBE4max and pLdCH-hyBE4max-LmajDBD plasmids. Plasmid libraries, in triplicate, were transfected into *L. mexicana* wildtype parasites. To determine the guide coverage (transfectants per guide), aliquots of replicates were subjected to 96-well plate cloning immediately after transfection. DNA was subsequently isolated 12 and 21 days after library transfection. Plasmid libraries and isolated DNA (all marked with red asterisks) were amplified and guide abundances in each sample and replicate were determined by Illumina sequencing. (**B**) Illumina sequencing data of both library screens (top panel: pLdCH-hyBE4max-LmajDBD; bottom panel: pLdCH-hyBE4max) was analysed by *z*-scoring the ratio between each plasmid, day 12 and 21 replicate sample. Z-scores were plotted against the total of raw reads from either: (1) plasmid and day 12 sample (left panel), or (2) plasmid and day 21 sample (right panel). To identify data points that were significant different, a 0.8 confidence ellipse was calculated (black line). Error bars show standard deviation between triplicates. Identities of data points, each representing a different guide sequence, are indicated in the legend below (see main text for further explanation).

*Figure 7 continued on next page*

*Figure 7 continued*

The online version of this article includes the following figure supplement(s) for figure 7:

**Figure supplement 1.** Conceptualisation and testing of a *Leishmania*-optimised hyBE4max version.

**Figure supplement 2.** Guide representation in a plasmid library delivered loss-of-function screen.

apply for RNAi approaches, their application remains restricted to *Leishmania* species of the *Viannia* subgenus (*Lye et al., 2010*; *Ullu et al., 2004*). In addition, the RNAi efficacy in *Leishmania Viannia* species varies widely and has been shown to be gene-, species-, and even strain-specific, as well as to depend on the length of the dsRNA hairpin (*de Paiva et al., 2015*; *Lye et al., 2022*; *Brettmann et al., 2016*; *Kohl et al., 2018*).

On the contrary, the base editing efficacy was here also shown to be time dependent and to vary widely amongst genes, species, and chosen sgRNA sequences. This is not surprising, especially as variations in efficacy across sgRNAs that target the same gene and locus are a common problem in any CRISPR approach, including CRISPR bar-seq screens (*Labuhn et al., 2018*). It is also worth highlighting that by base editing produced knockdowns are technically truncations and open reading frames (ORFs) are not fully replaced. It is unknown whether mutant mRNAs harbouring these truncated ORFs have any additional effects on the parasites or whether they are quickly degraded. Degradation of mutant mRNAs in which a termination codon is present within the ORF would require a classical nonsense-mediated decay pathway. But it is unclear whether this pathway is functional in *Leishmania* and other kinetoplastids (*Clayton, 2019*; *Delhi et al., 2011*). Thus remaining N-terminal protein parts could be functional or even deleterious themselves, leading to false positive and negative results. However, this may also provide extra information about functional domains of the targeted protein and highlights that our tool can not only be used to create functional null mutants by inserting premature STOP codons but also to pursue targeted mutagenesis screens.

To facilitate large-scale base editing CRISPR screens, ssDNA oligo pools can be obtained commercially and used to generate sgRNA libraries. The generation of CRISPR libraries is straight forward and scalable, with some libraries containing as many as ~215,000 sgRNAs (*Horlbeck et al., 2016*). Guides are designed to be gene specific and have therefore a much greater specificity than RNAi plasmid libraries that are typically comprised of randomly sheared genomic DNA, such as those previously used for *T. brucei* (*Alsford et al., 2011*; *Morris et al., 2002*). Cloning strategies to generate sgRNA plasmid libraries typically involve first the amplification of ssDNA oligo libraries and then golden gate assembly (using Type IIS restrictions enzymes) of amplicons into a target vector. Since the vector we here use for base editing is compatible with this cloning strategy, we believe that our base editing method could be scaled up to facilitate genome-wide screens in future studies. The ability of just sharing targeted or potentially even genome-wide plasmid libraries and applying these to different research questions in various *Leishmania* species in many different laboratories around the globe represents a major improvement to previous methods and overcomes challenges associated with bar-seq screens, such as the LeishGEM project (as outlined in our introduction).

## Possible improvements to base editing in *Leishmania*

What remains to be improved however, is the long culture time needed for some guides in order to reach high C to T conversion rates because prolonged guide targeting may cause unwanted side effects, such as off-target mutations (*Grünewald et al., 2019*; *Zhou et al., 2019*). We did not assess possible off-target mutations on a genome-wide scale in this study and this remains to be thoroughly evaluated in the future. However, apart from one guide (PF16-1 in *L. donovani*; *Supplementary file 2*), which resulted in editing within 6nt range of the guide sequence, we did not detect any editing outside of the 20nt guide sequence even 33 days post transfection when analysing the entire Sanger sequencing range of all our targeted genes (*Figures 3A, B, 5A–C, and 6D, E* and *Supplementary file 2*).

Editing time could be reduced by fine-tuning sgRNA and CBE expression, especially through the use of species-specific promoters, synthetic promoters, modified T7 RNAP promoters and stable integration of expression constructs. This could also help to minimise possible toxic effects on growth, as observed in *L. donovani* and *L. infantum* (*Figure 2—figure supplement 2B*). Integrating sgRNA constructs would also ensure that sgRNAs are not lost during *in vivo* screening and that just one

sgRNA per transfectant is expressed, thereby reducing the risk of horizontal plasmid transfer between parasites throughout a screen. Multiple sgRNAs per transfectant could theoretically cause multi-gene knockout effects. However, this risk seems to be low as we did not detect more than two sgRNAs per transfectant and as those two guides seemed to be randomly mixed (*Figure 7—figure supplement 2B*). In addition, integrating sgRNA constructs would reduce possible variations in the episomal copy number, which *Leishmania* parasites are known to vary (*Ubeda et al., 2014*). Although episomal transfection allows also to adjust protein expression by increasing the concentration of selection drugs (*Roberts et al., 2007*), our attempts to increase editing rates by raising drug concentrations were not successful (*Figure 4C, D*, *Figure 4—figure supplement 4*).

To reduce editing time, it may be also preferable to target the template strand (−) over the coding strand (+). Since mutations on the coding strand have no immediate effect on the transcript, they first need to be incorporated on the template strand (−) during DNA replication. Hence, targeting the coding strand leads to a slower increase in editing over time and this was also repeatedly reflected in our data (*Figure 2D*, *Figure 2—figure supplement 2A*, and *Figure 7—figure supplement 1C*). We therefore have adjusted our scoring criteria for ranking guides on our primer design platform http://www.leishbaseedit.net/. In addition, to minimise the possibilities for non-STOP codon mutations to become dominant our guide ranking could be further modified in the future to prioritise guides that have as few as possible cytosine bases. Although we have not measured this yet, the expectation would be that guides with fewer cytosine bases within the editing window would result in higher rates of successful STOP codon transformations. This is especially desirable when targeting essential genes, as the chances of non-deleterious mutations becoming dominant would be reduced.

Lastly, modifying the CBE version through codon optimisation, addition of species-specific NLS or exchange of deaminase and UGI domains, could further increase the editing rate in *Leishmania*. We show that our *Leishmania*-optimised hyBE4max CBE version increases the editing efficiency (*Figure 7B* and *Figure 7—figure supplement 1C*). However, it remains to be explored whether also fidelity and editing window size is affected by this modification, as for example hyBE4max adaptations in rise are known to have expanded the editing windows (*Wei et al., 2022*). While especially our improved CBE version allowed to efficiently induce a knockdown of CRK3, guides targeting IFT88 were not significantly depleted in our small-scale library screen (*Figure 7B*). Although IFT88 is not an essential gene, we would have also expected a decrease in guide abundance over time as IFT88 has been shown to be required for normal promastigote proliferation (*Beneke et al., 2019*). It is important to note that we prioritised universal sgRNAs over the highest scoring guides to target IFT88 in multiple species in this study. Testing these four IFT88 guides individually, we only found one to be effective in *L. infantum*, while in *L. major* no measurable effect, even 42 days post transfection, was observed (*Figure 4A–D*). Meanwhile, we did not individually test these four IFT88 targeting guides in *L. mexicana*. It is therefore possible that by picking higher scoring guides, a depletion of IFT88 guides from the pool could have been achieved. It is also worth highlighting again that through base editing produced knockdowns are technically truncations with remaining N-terminal protein parts that could be functional. In addition, the exact timing of isolating DNA from pools might influence when a decrease of parasites with a deleterious mutation is detectable. As pointed out above, this may differ between guides, species, and genes. Thus, to detect the whole range of mutant phenotypes, such as IFT88, a tighter collection window of isolating DNA from pools may be needed. Nevertheless, we demonstrate through targeting of CRK3 that cells with an introduced premature STOP codon in an essential gene can be significantly depleted from a library with our approach (*Figure 7B*).

## Conclusion

To date, a functional dissection of the vast majority of *Leishmania* genes remains to be performed and a lack of suitable tools limits the ability of the field to pursue large-scale loss-of-function screens. There is an urgent need to develop a high-throughput technology that allows: (1) to identify genes that are essential in any given environment, (2) to study multi-copy genes, and (3) to generate loss-of-function libraries through transfection of pooled plasmids rather than by pooling individually generated mutants. Here, we report a CRISPR/Cas9 CBE toolbox for *Leishmania* that fulfils these criteria. LeishBASEedit allows to efficiently transform codons into STOP codons, thereby enabling the functional disruption of single- and multi-copy genes without requiring DSBs, HR, donor DNA, or isolation of clones. We demonstrate that this tool can be used for drug resistance screening, to target

essential genes and to pursue loss-of-function screens in a pooled plasmid library transfection format. We therefore believe that this method will enable large-scale loss-of-function screens in *Leishmania* in many laboratories around the globe, simply by just sharing plasmid libraries. This will potentially even allow for genome-wide screens under different conditions in various clinical isolates and across multiple *Leishmania* species without limitations due to extreme cases of gene copy numbers and/or aneuploidy or lack of RNAi components.

# Materials and methods

### Key resources table

| Reagent type (species) or resource | Designation | Source or reference | Identifiers | Additional information |
|---|---|---|---|---|
| Cell line (*Leishmania mexicana*) | *L. mexicana* wildtype | Eva Gluenz laboratory | WHO strain MNYC/BZ/62/M379 | Used TriTrypDB (release 59, *Aslett et al., 2010*) reference annotation: *L. mexicana* MHOMGT2001U1103 |
| Cell line (*Leishmania major*) | *L. major* wildtype | Eva Gluenz laboratory | Strain Friedlin | Used TriTrypDB (release 59, *Aslett et al., 2010*) reference annotation: *L. major* Friedlin |
| Cell line (*Leishmania donovani*) | *L. donovani* wildtype | Joachim Clos laboratory, *Decuypere et al., 2005* | Strain BPK190 | Used TriTrypDB (release 59, *Aslett et al., 2010*) reference annotation: *L. donovani* BPK282A1 |
| Cell line (*Leishmania infantum*) | *L. infantum* wildtype | Joachim Clos laboratory, *Sulahian et al., 1997* | Strain MHOM/FR/91/LEM2259 | Used TriTrypDB (release 59, *Aslett et al., 2010*) reference annotation: *L. infantum* JPCM5 |
| Recombinant DNA reagent | hyBE4max | Addgene, *Zhang et al., 2020* | #157942 | |
| Recombinant DNA reagent | AncBE4max | Addgene, *Koblan et al., 2018* | #112100 | |
| Recombinant DNA reagent | pTB007 | *Beneke et al., 2017* | | |
| Recombinant DNA reagent | pPLOT Puro | *Beneke et al., 2017* | | |
| Recombinant DNA reagent | pLdCH | Addgene, *Zhang et al., 2017* | #84291 | |
| Recombinant DNA reagent | pTB007-hyBE4max | This study | | See description under 'Construction of CBE and sgRNA expression plasmids' |
| Recombinant DNA reagent | pTB007-AncBE4max | This study | | See description under 'Construction of CBE and sgRNA expression plasmids' |
| Recombinant DNA reagent | pLdCH-hyBE4max | This study | | See description under 'Construction of CBE and sgRNA expression plasmids' |
| Recombinant DNA reagent | pLdCH-hyBE4max-LmajDBD | This study | | See description under 'Construction of CBE and sgRNA expression plasmids' |
| Antibody | Guide-it Cas9 (rabbit polyclonal) | Takara | #632607 | Dilution for western blot 1:1000 |
| Antibody | PFR2 (mouse monoclonal) | *Kohl et al., 1999* | L8C4 | Dilution for western blot 1:10 |
| Chemical compound, drug | Miltefosine | Sigma | M5571 | |
| Chemical compound | Thiazolyl Blue Tetrazolium Bromide (MTT) | Sigma | M2128 | |

*Continued on next page*

*Continued*

| Reagent type (species) or resource | Designation | Source or reference | Identifiers | Additional information |
|---|---|---|---|---|
| Software | ICE | Synthego | https://ice.synthego.com/#/ | |
| Software | TriTrypDB (release 59) | *Aslett et al., 2010* | https://tritrypdb.org/tritrypdb/app | |
| Software | LeishBASEedit | This study | http://www.leishbaseedit.net/ | See description under 'Automated CBE guide design using LeishBASEedit' |
| Gene (*L. donovani* BPK282A1, *L. infantum* JPCM5, *L. major* Friedlin, *L. mexicana* MHOMGT2001U1103) | PF16 | TriTrypDB (release 59), *Aslett et al., 2010* | LdBPK_201450.1, LmjF.20.1400, LmxM.20.1400, LINF_200019300 | |
| Gene (*L. donovani* BPK282A1, *L. infantum* JPCM5, *L. major* Friedlin, *L. mexicana* MHOMGT2001U1103) | IFT88 | TriTrypDB (release 59), *Aslett et al., 2010* | LINF_270017700, LdBPK_271010.1, LmjF.27.1130, LmxM.27.1130 | |
| Gene (*L. donovani* BPK282A1) | MFT | TriTrypDB (release 59), *Aslett et al., 2010* | LdBPK_131590.1.1 | |
| Gene (*L. mexicana* MHOMGT2001U1103) | PFR2A, B, and C | TriTrypDB (release 59), *Aslett et al., 2010* | LmxM.16.1430.1 | |
| Gene (*L. mexicana* MHOMGT2001U1103) | CRK3 | TriTrypDB (release 59), *Aslett et al., 2010* | LmxM.36.0550.1 | |

## Cell culture

Promastigote-form *L. mexicana* (WHO strain MNYC/BZ/62/M379), *L. major* Friedlin, *L. donovani* (strain BPK190, *Decuypere et al., 2005*), and *L. infantum* (strain MHOM/FR/91/LEM2259, *Sulahian et al., 1997*) were grown at 28°C in M199 medium (Life Technologies) supplemented with 2.2 g/l NaHCO$_3$, 0.0025% haemin, 0.1 mM adenine hemisulfate, 1.2 µg/ml 6-biopterin, 40 mM 4-(2-hydroxyethyl) piperazine-1-ethanesulfonic acid pH 7.4, and 20% FCS (fetal calf serum). Media were supplemented with the relevant selection drugs: 40 or 400 µg/ml Hygromycin B, 40 µg/ml Puromycin Dihydrochloride, and 40 µg/ml G-418 Disulfate. The identity of each *Leishmania* species was confirmed by sequencing the LPG2 locus (*Akhoundi et al., 2017*; *Figure 2—figure supplement 1*) and cell lines were tested negative for mycoplasma contamination at Eurofins Genomics. Doubling times were determined by sub-culturing parasites repeatedly (over 5 days) to 10$^6$ cells/ml and measuring cell densities 24 hr later using a Coulter Counter Z2 particle counter (Beckman Coulter).

## Automated CBE guide design using LeishBASEedit

For design of sgRNA primers, coding sequences (CDS) and genome reference sequences were obtained from TriTrypDB (release 59) (*Aslett et al., 2010*) and processed using an in-house script (Source Code File 1, LeishBASEedit v1). First CRISPR-CBEI (*Yu et al., 2020*) was used to find all suitable sgRNAs that would transform arginine, tryptophan, or glutamine codons into STOP codons by cytosine to thymine conversion. Then, sgRNAs were filtered to select guides that would introduce a STOP codon within the first 50% of the CDS and were scored via a scoring matrix as follows: (1) one scoring point per additionally introduced STOP codon, (2) one scoring point if C to T edit is within 4–8 editing window, (3) one scoring point if C to T edit is within 4–10 editing window (guides with 4–8 editing window will have now 2 points in total), (4) one scoring point if C to T edit is within first 20% of CDS, and (5) one scoring point if C to T edit is within first 40% of CDS (guides within first 20% of CDS will have now 2 points in total) (the scoring matrix can be found as a table on http://www.leishbaseedit.net/). If the sgRNA score was equal, the guide closer to the START codon was ranked higher. In addition, sgRNAs targeting the template strand (−) were ranked higher than sgRNAs targeting the coding strand (+) (as further explained in the discussion). Finally, we also assessed the specificity of each sgRNA by counting the number of perfect alignments between guide target sequence (23 nt, including the protospacer adjacent motif NGG) and reference genome. We then transformed

all sgRNA target sequences to generate primers for cloning into pLdCH-hyBE4max and pLdCH-hyBE4max-LmajDBD plasmids, and uploaded CBE sgRNA design data for 64 different kinetoplastids to our new primer design resource http://www.leishbaseedit.net/.

For all sgRNAs used in this study (*Supplementary file 5*), we used the following reference annotations from TriTrypDB (release 59, *Aslett et al., 2010*): (1) for *L. donovani* (strain BPK190), we used the *L. donovani* BPK282A1 annotation, (2) for *L. infantum* (strain MHOM/FR/91/LEM2259), we used the *L. infantum* JPCM5 annotation, (3) for *L. major* Friedlin, we used the *L. major* Friedlin annotation, and (4) for *L. mexicana* (WHO strain MNYC/BZ/62/M379), we used the *L. mexicana* MHOMGT2001U1103 annotation.

## Construction of CBE and sgRNA expression plasmids

To generate pTB007-hyBE4max (*Figure 3—figure supplement 1A*) and pTB007-AncBE4max (*Figure 3—figure supplement 2A*) plasmids, hyBE4max (Addgene #157942, *Zhang et al., 2020*) and AncBE4max (Addgene #112100, *Koblan et al., 2018*) were amplified using primer 1006F and 1007R and cloned into SpeI and FseI sites of pTB007 (*Beneke et al., 2017*). Plasmid pLdCH-hyBE4max (*Figure 2A*) was generated by amplifying hyBE4max (Addgene #157942, *Zhang et al., 2020*) using primer 1008F and 1007R and cloning it into AgeI and FseI sites of pLdCH (Addgene #84291, *Zhang et al., 2017*). To construct pLdCH-hyBE4max-LmajDBD (*Figure 7—figure supplement 1B*), a fusion PCR construct was produced and cloned into pLdCH-hyBE4max, thereby replacing the human-derived Rad51 ssDBD. Briefly, three fragments were individually amplified: (1) C-terminal end of APOBEC-1 from the pLdCH-hyBE4max plasmid (primer 1084F_1stRXN and 1085Rfusion), (2) Rad51 ssDBD from *L. major* Friedlin genomic DNA (primer 1086F and 1087R), and (3) nCas9 from the pLdCH-hyBE4max plasmid (primer 1088Ffusion and 1014R). Then all three fragments were fused in a fusion PCR reaction using 1084F and 1007R as nested primers. Finally, the resulting fusion PCR construct was cloned into AvrII and FseI sites of pLdCH-hyBE4max.

To generate plasmids for T7 RNA polymerase (RNAP)-driven sgRNA transcription (namely pPLOT-T7GuideExpress, *Figure 3—figure supplement 1B* and *Figure 3—figure supplement 2B*), primers 1033F–1038F were individually mixed with primer 1039R and pLdCH plasmid DNA (Addgene #84291, *Zhang et al., 2017*) to amplify an sgRNA expression cassette, consisting of: (1) T7 RNAP promoter sequence, (2) the sgRNA target sequence, (3) the Cas9 scaffold, and (4) a hepatitis delta virus (HDV) ribozyme. Amplicons were then cloned into MluI and NsiI sites of a pPLOT-Puro plasmid (*Beneke et al., 2017*).

To clone sgRNA sequences into pLdCH-hyBE4max and pLdCH-hyBE4max-LmajDBD plasmids, we followed the guide cloning protocol described previously (*Shalem et al., 2014*; *Sanjana et al., 2014*) with minor modifications (*Supplementary file 6*, also downloadable from http://www.leishbaseedit.net/). Using this protocol we reached 95.8% cloning efficiency across all 26 cloned guides (*Supplementary file 5*) (46 out of 48 isolated bacteria colonies correctly cloned on first attempt).

Constructed plasmids (pTB007-hyBE4max, pTB007-AncBE4max, pLdCH-hyBE4max, and pLdCH-hyBE4max-LmajDBD) were Sanger sequenced, covering completely the CBE in each plasmid. Following sgRNA cloning, each pLdCH-hyBE4max or pLdCH-hyBE4max-LmajDBD plasmid was Sanger sequenced using primer 1011F, while each pPLOT-T7GuideExpress plasmid was Sanger sequenced using primer M13 rev-29.

All used primer sequences (*Supplementary file 5*) and plasmid maps (*Supplementary file 7*) can be found in the supplementary. Primers were all ordered as standard desalted oligos at 25 nmol scale (Sigma).

## Transfections

Parasites were transfected as described previously (*Beneke et al., 2017*; *Beneke and Gluenz, 2019*) in 1× Tb-BSF buffer (*Schumann Burkard et al., 2011*). Briefly, 5–10 µg of tdTomato expression construct (see description for 'tdTomato reporter assays' below) or plasmid DNA (circular for pLdCH-hyBE4max, pLdCH-hyBE4max-LmajDBD, and pPLOT-T7GuideExpress plasmids; PacI linearised for pTB007-hyBE4max and pTB007-AncBE4max) were diluted into 50 µl ultra-pure water and heat sterilised for 5 min at 95°C. Then, 50 µl sterilised DNA was mixed with 5 × 10^6 cells submerged in 200 µl of 1.25× Tb-BSF buffer (final transfection volume 250 µl with 1× Tb-BSF buffer) and transfected using one pulse with program X-001 on an Amaxa Nucleofector IIb (Lonza). Cells were recovered in M199

culture medium as specified above and 6–16 hr post transfection the required selection drug was added. A detailed protocol for transfecting CBE plasmids into *Leishmania* parasites can be found in the supplementary (*Supplementary file 8*) or on http://www.leishbaseedit.net/.

## TdTomato reporter assays

To set up tdTomato reporter assays, we first generated a pPLOT-Neo-tdTomato plasmid. The plasmid pTSARib-tdTomato (*Capewell et al., 2016*) was digested with HindIII and BamHI to isolate the tdTomato-coding sequence, which was then cloned into HindIII and BamHI sites of pPLOT-Neo (*Beneke et al., 2017*). Then a tdTomato expression construct was generated, consisting of: (1) a neomycin drug resistance expression cassette, (2) a tdTomato expression cassette, and (3) two surrounding homology flanks allowing for integration into the 18S rRNA locus. The construct was produced by fusion PCR as previously described (*Dean et al., 2015*). First, two fragments were individually amplified: (1) upstream and (2) downstream homology flanks from *L. major* Friedlin genomic DNA using primer 1000F/1001R and primer 1002F/1003R, respectively. Then both fragments were used to amplify the tdTomato expression construct from the pPLOT-Neo-tdTomato plasmid, using primer 1004F and 1005R as nested primers. Finally, PCR reactions were purified and transfected as described above. Following transfection into *L. mexicana*, *L. major*, *L. donovani*, and *L. infantum*, clones were isolated as previously described (*Beneke et al., 2017*) and screened using a FACSCalibur cell analyzer (BD) to select a clone with high tdTomato expression. Subsequently, the chosen *L. major* tdTomato-expressing clone was again transfected with PacI linearised pTB007-hyBE4max and pTB007-AncBE4max plasmid DNA. Clones were again isolated, expression of tdTomato confirmed and the fastest growing clone picked for the tdTomato reporter assay. We did not assess how many copies of the tdTomato expression cassette were integrated in selected tdTomato clones.

Chosen tdTomato clones were then transfected with pLdCH-hyBE4max and pLdCH-hyBE4max-LmajDBD plasmids, carrying tdTomato-specific sgRNAs. Meanwhile, tdTomato-pTB007-hyBE4max and tdTomato-pTB007-AncBE4max selected clones were transfected with pPLOT-T7GuideExpress plasmids, carrying also tdTomato-specific sgRNA target sequences. Non-clonal populations were analysed using a FACSCalibur cell analyzer (BD) several days post transfection (as indicated). FACS data were analysed using Flowing Software 2 (Turku Bioscience).

To prepare parasites for FACS analysis, 400–800 µl of a dense culture was pelleted at 800 × *g* for 5 min, washed once in phosphate-buffered saline (PBS, pH 7.4) and then resuspended in 200 µl PBS.

## Motility analysis and light microscopy

Motility analysis was performed as previously described in *Beneke et al., 2019* using the method from *Wheeler, 2017*. Mean speed was measured in duplicates 14 days post transfection and in triplicates 28 and 42 days post transfection. 5 µl of cell culture (cell density ~6 × 10$^6$ cells/ml) was placed on a glass slide in a 250-µm deep chamber covered with a 1.0-cover slip and imaged using darkfield illumination with a ×10 numerical aperture (NA) 0.3 objective and an Allied-Vision Pike F-505B camera on a Zeiss Axiophot microscope at the ambient temperature of 25–28°C. Tracks were processed using a Fiji (*Schindelin et al., 2012*) macro from *Wheeler, 2017* and further processed by creating histograms of velocity per tracked cell with the following bin categories: 0.01, 0.1, 0.3, 0.7, 1.0, 1.5, 3.0, 6.0, 8.0, 10.0, 12.0, 16.0, and 30.0 µm/s. Histogram data of replicates were then averaged and tested using the Cramér–von Mises criterion, allowing to identify which non-clonal mutant populations significantly developed a positively (right) skewed distribution due to the base editing (*Supplementary file 1*).

For light microscopy, *Leishmania* parasites were prepared as described previously (*Wheeler et al., 2015*) and imaged live using an inverted fluorescence microscope (DMI6000B, Leica) with a ×63 NA 1.30 glycerine immersion objective and a Leica DFC365 FX monochrome digital camera at the ambient temperature of 25–28°C.

## Western blots

Cas9 and PFR2 protein was detected on western blots using the Guide-it Cas9 (Takara, 632607) and L8C4 (*Kohl et al., 1999*) (a gift from K. Gull, University of Oxford) antibodies, respectively. Briefly, *Leishmania* promastigotes were washed in ice-cold PBS twice containing 1× Halt Protease Inhibitor Cocktail (Thermo Fisher) and then heated in 1× Laemmli buffer for 10 min at 60°C. 2.5 × 10$^6$ cell equivalents of *Leishmania* promastigotes were loaded and subjected to electrophoresis on 6% (for Cas9)

or 10% (for PFR2) sodium dodecyl sulfate (SDS)–polyacrylamide gels. Samples were transferred to nitrocellulose membranes (GE Healthcare) and incubated in blocking buffer (PBS with 0.05% [wt/vol] Tween 20 [PBST] with 5% milk for Cas9 and 1% milk for L8C4) for 1 hr. Then, membranes were incubated with primary antibodies (diluted in blocking buffer [Cas9 1:1000, L8C4 1:10]) for 1 hr, washed in PBST three times and incubated with secondary antibodies (IRDye LI-COR, 926-32211 for Cas9, 925-32210 for L8C4, both diluted 1:10,000 in PBST) for 1 hr. Membranes were washed again in PBST three times and imaged using a LI-COR Odyssey CLx.

## Miltefosine drug resistance and MTT viability assays

Transfected and non-transfected *L. donovani* parasites were treated for 48 hr with 20 µM miltefosine (pre-treated parasites) from 16 days post transfection onwards. Then cells were washed in PBS twice and cultured for another 72 hr. In the following, pre-treated and non-pre-treated parasites were aliquoted into a 96-well plate to measure cell viability in triplicates when incubated with different doses of miltefosine (*Figure 6A*). For each 96-well plate well, $5 \times 10^4$ parasites were resuspended in 85 µl cell culture medium and mixed with 5 µl of sterile ultra-pure water or miltefosine solution (Sigma M5571, dissolved in ultra-pure water at 10 mM) to a final concentration of either 5, 20, 50, 80, and 400 µM. Cells were cultured for 48 hr and then 10 µl of Thiazolyl Blue Tetrazolium Bromide solution (MTT; Sigma M2128, dissolved in PBS at 5 mg/ml and 0.2 µm sterile filtered) was added to measure cell viability and cells were incubated at 28°C in the dark for 3 hr. Subsequently, 100 µl of 10% SDS solution (dissolved in 0.01 M HCl) was added and plates were incubated at 37°C in the dark overnight (not longer than 18 hr) (*Tada et al., 1986*). Finally, the absorbance was read at 570 nm using a Tecan Infinite M200 Plate Reader (Tecan Life Sciences).

For analysis, absorbance reads from the media control were subtracted from the absorbance read of each data point and then the ratio between wells with no added miltefosine and each different miltefosine concentration was calculated. The significance was calculated by using a one-way analysis of variance test with post hoc Tukey honestly significant difference (*Supplementary file 3*).

## DNA isolation, Sanger sequencing, and measurement of editing rates

Following transfection, DNA was isolated as previously described (*Rotureau et al., 2005*). Briefly, 400–800 µl of a dense culture was pelleted, resuspended in 100 µl lysis buffer (10 mM Tris–HCl at pH 8.0, 5 mM EDTA (ethylenediaminetetraacetic acid), 0.5% SDS, 200 mM NaCl, and 100 µg/ml proteinase K) and incubated for 30 min at 65°C. Then, 250 µl of 100% EtOH was added and samples were centrifuged at $17,000 \times g$ for 30 min (at 4°C). Supernatant was aspirated and DNA resuspended in 50 µl of ultra-pure water. To amplify amplicons for Sanger sequencing, 1 µl of DNA was added to a 50 µl PCR reaction. Amplicons were generated using two guide target locus spanning primers and were either directly purified or first ran on a 2% agarose gel and then purified following band excision from the gel. Depending on the location of the guide, the forward or reverse primer was then used for Sanger sequencing at Eurofins Genomics. Primers for amplification and Sanger sequencing are listed in *Supplementary file 5*. To measure the editing rate, sequencing reads were analysed and normalised using ICE (Synthego), allowing to compare the 'before-editing' read (control sample) with the 'after-editing' read (edited sample) by overlapping normalised peak heights at aligned positions. The resulting discordance between both reads was then mapped to guide positions and plotted as editing rate.

## Generation of sgRNA CBE plasmid libraries

To generate plasmid libraries, we cloned a pool of sgRNA sequences into pLdCH-hyBE4max and pLdCH-hyBE4max-LmajDBD CBE plasmids. First, plasmid DNA was linearised and dephosphorylated by incubating the following mixture at 37°C overnight: 5 µg CBE plasmid, 2 µl BpiI (BbsI) (Thermo Fisher, ER1011), 2 µl FastAP (Thermo Fisher, EF0651), and 3 µl 10× Tango Buffer (Thermo Fisher) (30 µl total volume with ultra-pure water). Then, sgRNA oligo pairs (*Supplementary file 5*) were all individually annealed and phosphorylated. 1 µl oligo 1 (100 µM), 1 µl oligo 2 (100 µM), 1 µl T4 DNA Ligation Buffer (Thermo Fisher), 0.5 µl T4-Polynukleotid-Kinase (Thermo Fisher, EK0031), and 6.5 µl ultra-pure water were mixed, then phosphorylated at 37°C for 30 and 5 min at 95°C, and finally annealed by ramping down from 95 to 25°C at 5°C/min. All 26 individually annealed and phosphorylated primer pairs were then equally mixed (10 µl of each oligo pair) and the resulting pool was diluted 1 in 200 by

mixing 5 µl of the oligo pool with 995 µl of ultra-pure water. Then the ligation of the library was set up by mixing 200 ng of PCR purified linearised and dephosphorylated plasmid DNA, 4 µl diluted oligo pool, 4 µl T4 DNA Ligation Buffer (Thermo Fisher) and 2 µl T4 DNA Ligase (Thermo Fisher, EL0014) (40 µl total volume with ultra-pure water). Ligations were incubated at 37°C for 2 hr, then mixed with 200 µl TOP10 competent cells, incubated for 30 min on ice, heat shocked at 42°C for 45 s and finally plated on two 10 cm LB agar plates (100 µl each plate), containing 50 µg/ml ampicillin. A negative ligation control without addition of the diluted oligo pool was processed alongside. Following over-night incubation at 37°C, transformation efficiency for all libraries was assessed to ensure at least 30× guide coverage (at least 820 colonies, assuming a 95.8% cloning efficiency as stated above). Typically, 30–200× guide coverage is used for generation of sgRNA libraries (*Sanjana et al., 2014*). Then all colonies were scrapped from the plate, collected in a 70-ml LB culture (containing 100 µg/ml ampi-cillin), incubated again overnight and processed the next day using a Macherey-Nagel NucleoBond PC Midi 100 kit (740573), yielding purified plasmid library DNA.

## Plasmid library delivered loss-of-function screen

To set up a plasmid library delivered loss-of-function screen, *L. mexicana* promastigotes were trans-fected as described above but using $1.3 \times 10^7$ cells per transfection and 100,000 plasmid molecules per cell (15.6 µg DNA of the pLdCH-hyBE4max plasmid library and 15.8 µg DNA of the pLdCH-hyBE4max-LmajDBD plasmid library). Following the transfection, cells were resuspended in 8.5 ml of culture medium. Immediately after resuspending cells, 750, 75, and 7.5 µl of the suspension were further diluted in culture medium (to 4 ml total volume) and each dilution was plated on 32 wells of a 96-well plate (100 µl per well). Then, 6 hr post transfection, hygromycin B was diluted in culture medium and added to plates and each transfected population (final volume: plates 200 µl, flasks 10 ml [final concentration 40 µg/ml]). Each library was transfected in triplicates and each of those replicates was diluted to assess the transfection efficiency.

Following transfection, populations were sub-cultured every 2–4 days from 7 days post transfection onwards. At least 13,000 cells per passage were transferred each time, ensuring a representation of 500 transfectants per guide on average throughout the screen. Plates were assessed 14 days post transfection by counting the number of isolated clones, allowing to calculate the number of transfec-tants per $10^6$, $10^5$, and $10^4$ cells, respectively, for each dilution.

## Next generation sequencing and screen analysis

DNA was isolated as described above from at least $5 \times 10^6$ cells at 12 and 21 days post transfection. Then, 1 µg DNA from both time points and 10 ng DNA from plasmid libraries was amplified using Phusion High–Fidelity DNA Polymerase (Thermo Fisher, F530S, GC Buffer protocol) and standard desalted p5 and p7 primers (Sigma, *Supplementary file 5*), containing inline barcodes for multi-plexing and partial adapters for Illumina sequencing. Inline barcodes had a hamming distance of at least 4nt. To avoid over-amplification, sample days 12 and 21 were both amplified using 28 PCR cycles, while plasmid samples were amplified using only 18 PCR cycles (ideal number of PCR cycles was assessed by testing 16, 18, 20, and 23 PCR cycles for plasmid samples and 25, 28, 30, and 32 PCR cycles for sample days 12 and 21). Amplicons were pooled (6 amplicons per pool), each pool purified (Macherey-Nagel NucleoSpin Gel and PCR Clean-up Kit) and send to GENEWIZ Germany GmbH for Illumina sequencing (Amplicon-EZ Sequencing Service, 250 bp paired-end sequencing).

To analyse sequencing data, cutadapt (*Martin, 2022*) was used to trim FASTQ sequencing reads, to generate a reverse complement read of the forward read and then to de-multiplex inline barcoded pools (using a 0.15 error rate). De-multiplexed forward and reverse reads were then analysed using MAGeCK (*Li et al., 2014*) and guide sequences per sample were counted and normalised. Raw guide counts were used to confirm the unity of each library, highlighting that guide PF16-2 and MFT-2 were underrepresented in both plasmid libraries. Therefore, both were excluded in the further analysis, leaving 24 guides for analysis in each library. Normalised forward and reverse reads were then aver-aged and the ratio between each plasmid, day 12 and 21 sample was calculated. To calculate ratios for

zero reads, a pseudo read value of 1 was inserted. Finally, ratios were z-scored and a 0.8 confidence ellipse was calculated (*Supplementary file 4*).

## Acknowledgements

We thank Eva Gluenz for *L. mexicana* (WHO strain MNYC/BZ/62/M379) and *L. major* Friedlin, Joachim Clos for *L. donovani* (strain BPK190, *Decuypere et al., 2005*) and *L. infantum* (strain MHOM/FR/91/LEM2259, *Sulahian et al., 1997*), David Liu for AncBE4max (Addgene #112100, *Koblan et al., 2018*), Dali Li for hyBE4max (Addgene #157942, *Zhang et al., 2020*), and Greg Matlashewski for pLdCH (Addgene #84291, *Zhang et al., 2017*). The L8C4 PFR2 antibody (*Kohl et al., 1999*) was a gift from Keith Gull. We thank Eva Gluenz, Andreia Albuquerque-Wendt, and Brooke Morriswood for helpful comments on this manuscript, as well as all members of the Janzen, Morriswood, Kramer, Alsheimer und Engstler groups for preparation of general laboratory reagents, set up of laboratory infrastructures and helpful discussions. Tom Beneke is funded by an EMBO Postdoctoral Fellowship (ALTF 727-2021). Markus Engstler is funded by the Deutsche Forschungsgemeinschaft (DFG) grants EN305, SPP1726, SPP2332, and GRK2157, the German-Israeli Foundation for Scientific Research and Development (grant I-473-416.13/2018) and the Bundesministerium für Bildung und Forschung (NUM Organostrat).

## Additional information

### Funding

| Funder | Grant reference number | Author |
|---|---|---|
| European Molecular Biology Organization | ALTF 727-2021 | Tom Beneke |
| Deutsche Forschungsgemeinschaft | EN305 | Markus Engstler |
| Deutsche Forschungsgemeinschaft | SPP1726 | Markus Engstler |
| Deutsche Forschungsgemeinschaft | SPP2332 | Markus Engstler |
| Deutsche Forschungsgemeinschaft | GRK2157 | Markus Engstler |
| German-Israeli Foundation for Scientific Research and Development | I-473-416.13/2018 | Markus Engstler |
| Bundesministerium für Bildung und Forschung | NUM Organostrat | Markus Engstler |

The funders had no role in study design, data collection, and interpretation, or the decision to submit the work for publication.

### Author contributions

Markus Engstler, Resources, Funding acquisition, Writing - review and editing; Tom Beneke, Conceptualization, Resources, Data curation, Software, Formal analysis, Supervision, Funding acquisition, Validation, Investigation, Visualization, Methodology, Writing - original draft, Project administration, Writing - review and editing

### Author ORCIDs

Tom Beneke (iD) http://orcid.org/0000-0001-9117-2649

### Decision letter and Author response

Decision letter https://doi.org/10.7554/eLife.85605.sa1
Author response https://doi.org/10.7554/eLife.85605.sa2

## Additional files

### Supplementary files

• Supplementary file 1. Motility data. Sheet contains histogram data of mean velocity of tracked cells (average values given). STDEV: standard deviation between duplicates (14 days post transfection) and triplicates (28 and 42 days post transfection).

• Supplementary file 2. Sanger sequencing trace plots alignments following ICE analysis. *L. donovani*, *L. infantum*, *L. mexicana*, and *L. major* wildtype parasites were transfected with pLdCH-hyBE4max-sgRNA expression plasmids, targeting PF16, MFT, PFR2, and IFT88 with four guides each (see main text description). Trace plots before transfection (control sample) and 28 days after transfection (edited sample) were aligned using ICE (Syntheco). Horizontal black line: 20nt guide target sequence. Horizontal red dotted line: PAM sequence. Vertical grey dotted line: nCas9 (D10A) DNA single-strand break (nick) position.

• Supplementary file 3. MTT cell viability data. Sheet contains calculated ratios (relative growth) between 'no drug control' and different doses of miltefosine (average values given). STDEV: standard deviation between triplicates.

• Supplementary file 4. NGS (next generation sequencing) analysis of plasmid library delivered loss-of-function screen. First sheet: total of raw guide sequence counts. Second sheet: normalised counts and *z*-scored ratios between plasmid and day 12 sample or plasmid and day 21 sample. Guide sequences of PF16-2 and MFT-2 were underrepresented in both plasmid libraries and are therefore greyed out.

• Supplementary file 5. Primers used in this study. Primers are separated in multiple sheets, including primers for: (1) generation of tdTomato expression constructs, (2) construction of cytosine base editor (CBE) plasmids, (3) cloning sgRNA target sequences into expression vectors, (4) species typing, (5) plasmid validation, (6) measuring mutation rates following base editing, and (7) Illumina sequencing.

• Supplementary file 6. Guide cloning protocol. A step-by-step protocol to anneal oligos and clone sgRNAs into pLdCH-hyBE4max and pLdCH-hyBE4max-LmajDBD.

• Supplementary file 7. Plasmid maps. Genbank files of cytosine base editor (CBE) plasmids produced in this study, including: pLdCH_AncBE4max, pLdCH-hyBE4max, pLdCH-hyBE4max-LmajDBD, and pTB007-hyBE4max

• Supplementary file 8. Base editor guide plasmid transfection protocol. A step-by-step protocol to transfect pLdCH-hyBE4max and pLdCH-hyBE4max-LmajDBD guide expression plasmids into *Leishmania* parasites.

• MDAR checklist

• Source code 1. LeishBASEedit primer design script. 'LeishBASEedit_v1.sh' is a bash script that allows to design and score cytosine base editor (CBE) guides to introduce STOP codons within the first 50% of any given open reading frame (ORF). The 'hyBE4max_specifics.csv' file is required to define the base editing window. Reference and CDS test files are also provided for setting up LeishBASEedit locally. Instructions and dependencies can be found in the README part of the script. The LeishBASEedit primer design output for 64 different kinetoplastids (TriTrypDB release 59) was deposited on http://www.leishbaseedit.net/ (open source).

• Source data 1. Raw Ponceau stain TIF images. Leishmania promastigote protein samples were loaded and subjected to electrophoresis on SDS-polyacrylamide gels and then transferred to nitrocellulose membranes. Membranes were then stained with Ponceau and scanned. This is the raw TIF file of this scan. File "Source data 1 (labels)" is the same file but with identities of protein samples highlighted. Details for samples can be found in legends of *Figure 2—figure supplement 2* (top panel), *Figure 4-figure supplement 4* (middle panel) and *Figure 5-figure supplement 1* (bottom panel).

• Source data 2. Raw LI-COR Odyssey CLx TIF images. Nitrocellulose membranes with Leishmania promastigote protein samples were subject to western blots and imaged using a LI-COR Odyssey CLx. This is the raw TIF file of this scan. File "Source data 2 (labels)" is the same file but with identities of protein samples highlighted. Details for samples can be found in legends of *Figure 2-figure supplement 2* (bottom-left panel), *Figure 4-figure supplement 4* (bottom-right panel) and *Figure 5-figure supplement 1* (top panel).

## Data availability

All data generated or analysed during this study are included in this manuscript and supplementary file. LeishBASEedit is an open-source primer design tool available under http://www.leishbaseedit. net/. The Source Code files for LeishBASEedit are included in the supporting files.

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
