## [Editor Report]

Leishmanias cause a wide spectrum of diseases in many animals: in humans, symptoms range from disfiguring skin lesions to lethal visceral infections. Knowledge of gene function has been limited by aneuploidy, as well as the absence of RNAi (in most but not all species) and of non-homologous DNA end joining (which limits the utility of Crispr-Cas systems). In this paper, the authors describe the successful testing and application of a new method for functional screens in Leishmania, in which targeted cytosine base editing is used to introduce premature stop codons within open reading frames. The advantages and disadvantages of this approach, in comparison with others, are described in a balanced and careful way. This is a very important new tool for researchers in Leishmania, and might also serve as a model for other poorly tractable species.

---

## [Decision Letter]

**Decision letter after peer review:**

Thank you for submitting your article "Introducing the CRISPR/Cas9 cytosine base editor toolbox 'LeishBASEedit' – Gene editing and high-throughput screening in Leishmania without requiring DNA double-strand breaks, homologous recombination or donor DNA" for consideration by *eLife*. Your article has been reviewed by 2 peer reviewers, including Christine Clayton as the Reviewing Editor and Reviewer #1, and the evaluation has been overseen by Dominique Soldati-Favre as the Senior Editor.

Essential revisions:

1) The main request is to be more balanced, up-front, about the pros and cons of this approach relative to LeishGEM and – even though these are currently "only" at the preprint stage – RNAi screens.

2) Minor changes as requested in the detailed comments – more details in the Methods, for example.

*Reviewer #1 (Recommendations for the authors):*

The following is a list of pros and cons for this method that I could think of. The other experts (who are Leishmaniacs) may have more useful input; all methods have a variety of possibilities for false-negative results.

Pros/possibilities:

a) Screening a particular gene set before LeishGEM results are available?

b) Screening genes that are present in L. donovani or L. infantum, but absent in L. mexicana/braziliensis – or which might have different functions. (At present it works too poorly in L. major to be useful, so the authors' claim to have established the method "in four Leishmania species" is an over-statement.)

c) Editing multi-copy genes (although if the mutation is lethal all copies might still not be edited).

Cons for this method:

1) Introducing a premature stop codon may not disrupt protein function (or may cause some other unexpected effect). Since there is probably no classical nonsense-mediated decay (if Leishmania is like *T. brucei*), one cannot expect the mRNA to be degraded – which might otherwise enhance the effects of the stop codon. In the case of proteins that interact with others, the production of truncated proteins might actually be more deleterious that deleting a gene entirely, giving possible false-positive results.

2) Culturing for at least 2 weeks – and often longer – was required to get high percentages of mutants. How does this compare with the other methods?

3) Using pools (which is what this tool is supposed to be good for), any cells that harbour deleterious mutations would be competed out so no mutants would be obtained, and there would be strong selection for cells in which the guide RNA had been lost from the episome, or the episome otherwise rendered harmless, giving false negative results. This is compounded by the problem that not all guides work.

4) It is still necessary to synthesise each guide separately, even if they can be cloned and transfected in bulk – describe how this would be done, and compare with the Gateway method for the RNAi. (Although some guides might be lost /underrepresented after cloning, the resulting variability would be no worse than the variability in representation in RNAi libraries and would be controlled for during screens.)

*Reviewer #2 (Recommendations for the authors):*

Improvements to the methods, in places, need to express cell numbers in the correct format, and the same with H20.

Line 436 – would read better as 'the highest expression of hyBE4mas was seen in'

Figure 4 – error bars were difficult to see and the legend didn't state what they represented. Plus what do the percentages mean on graphs? Need a bit more detail in this figure legend.

Line 482 – a little explanation for why this result is expected would be useful.

Have the authors analysed the effectiveness of the guide based on whether the only mutation within the guide would result in the generation of a STOP versus those that offer multiple mutation options?

In figure S11A could the authors highlight the exact sequence of the Lmajor ssDBD they used to generate the modified plasmid?

The authors could include a discussion of the RNAi-based approaches for L. braziliensis in comparison to other methodologies.

---

## [Author Response]

Reviewer #1 (Recommendations for the authors):The following is a list of pros and cons for this method that I could think of. The other experts (who are Leishmaniacs) may have more useful input; all methods have a variety of possibilities for false-negative results.Pros/possibilities:a) Screening a particular gene set before LeishGEM results are available?

We did not develop LeishBASEedit to screen particular gene sets before LeishGEM does or to compete with this project. Instead it was developed to generate more easily mutant libraries for all sort of screening applications and use these in all *Leishmania* species without facing limitations due to the lack of RNAi activity or due to the presence of extreme cases of aneuploidy. The real advantage of our here developed method is that plasmid libraries can be easily shared around the globe with many different labs without requiring a modified cell line beforehand. We have clarified this point in our revised discussion (line 429 – 468).

b) Screening genes that are present in L. donovani or L. infantum, but absent in L. mexicana/braziliensis – or which might have different functions. (At present it works too poorly in L. major to be useful, so the authors' claim to have established the method "in four Leishmania species" is an over-statement.)

Screening genes that are present in one species and not in the other would be one possibility. However, since gene copy numbers greatly vary between *Leishmania* species and many single copy genes vary in their ORF sequence, it is also attractive to pursue comparative genome-wide screens in all *Leishmania* species. Performing the LeishGEM project in multiple *Leishmania* species would be unfeasible at present but feasible with our base editing method (see revised introduction [line 36 – 73] and discussion [line 429 – 468]).

We also don’t think that claiming we have established base editing in *L. major* is an over-statement. Due to the *L. donovani*-derived rRNA promoter in our expression construct there is of course room for improvement, which we highlight in our discussion (line 497 – 535). However, our experiments in *L. major*, including the reporter assays and the targeting of PF16, clearly show that it works (see Figure 2, 3 and 4).

c) Editing multi-copy genes (although if the mutation is lethal all copies might still not be edited).

We agree with this comment and made this argument already in our original manuscript (line 469 – 472). For clarity, we reiterate this point now in our revised version (line 430 – 433 and 570 – 572).

Cons for this method:1) Introducing a premature stop codon may not disrupt protein function (or may cause some other unexpected effect). Since there is probably no classical nonsense-mediated decay (if Leishmania is like *T. brucei*), one cannot expect the mRNA to be degraded – which might otherwise enhance the effects of the stop codon. In the case of proteins that interact with others, the production of truncated proteins might actually be more deleterious that deleting a gene entirely, giving possible false-positive results.

We added this comment to our revised manuscript (line 445 – 454).

2) Culturing for at least 2 weeks – and often longer – was required to get high percentages of mutants. How does this compare with the other methods?

Culturing for at least 2 weeks was not required to get high percentages of mutants but to recover cells from transfections. For our here developed base editing approach, cells are transfected with a plasmid that expresses the base editor, the sgRNA and a drug resistance gene. In order to edit a GOI, cells need to be selected for this plasmid. Selection of drug resistant cells takes between 7 – 10 days (depending on the species). In the following the cells need to be sub-cultured before DNA can be extracted to assess the base editing rate (to dilute out cell debris from the selection). Hence 14 days was just the earliest time point we could look at simultaneously across all species. It is very likely that the actual editing happens much earlier and immediately after transfection if the guide is efficient enough. Therefore, some guides result in 100% editing rate at the earliest time point. However, it is true that some guides then needed longer in culture in order to be efficient. This is due to differences in guide efficiencies, which can be observed for various CRISPR approaches, as commented in our discussion (line 441 – 444). In addition, for base editing the efficiency and editing time will depend on the strand that is being targeted (line 526 – 531).

3) Using pools (which is what this tool is supposed to be good for), any cells that harbour deleterious mutations would be competed out so no mutants would be obtained, and there would be strong selection for cells in which the guide RNA had been lost from the episome, or the episome otherwise rendered harmless, giving false negative results. This is compounded by the problem that not all guides work.

We agree, the strength of this tool lies within its compatibility for loss-of-function screens via plasmid pool delivery. Following a plasmid pool transfection, cells that harbour deleterious mutations will be outcompeted from the pool as shown in our proof of principle CRK3 screen (Figure 7). Some guides will not be effective, just like some guides for the LeishGEdit method will not be functional or like some RNAi hairpins have no effect. This is the reason why multiple guides in parallel are being tested and why large-scale CRIPSR libraries usually target genes with 3-10 guides. So we do not think “that not all guides work” is a problem specific for base editing.

In addition, we have added that our method is not only useful for using plasmid pools but may be also used for other applications, such as targeted mutagenesis experiments (line 452 – 454).

4) It is still necessary to synthesise each guide separately, even if they can be cloned and transfected in bulk – describe how this would be done, and compare with the Gateway method for the RNAi. (Although some guides might be lost /underrepresented after cloning, the resulting variability would be no worse than the variability in representation in RNAi libraries and would be controlled for during screens.)

We have addressed this comment in our revised discussion (line 455 – 468).

Reviewer #2 (Recommendations for the authors):Improvements to the methods, in places, need to express cell numbers in the correct format, and the same with H20.

Many thanks for these suggestions. We improved the method section as suggested.

Line 436 – would read better as 'the highest expression of hyBE4mas was seen in'

We have changed the sentence as suggested (line 253 – 255).

Figure 4 – error bars were difficult to see and the legend didn't state what they represented. Plus what do the percentages mean on graphs? Need a bit more detail in this figure legend.

We have changed the colour of the error bars and updated the legend of figure 4 as suggested.

Line 482 – a little explanation for why this result is expected would be useful.

As the reviewer highlighted himself, deleterious mutations that result in slower growth will be outcompeted by cells in which a non-deleterious mutation has occurred. We have stated that the complete deletion of IFT88 in *Leishmania mexicana* has been shown to have reduced doubling time (Beneke et al., PLoS Pathogens 2019) and are therefore most likely outcompeted from the pool (line 293 – 298 and 537 – 541).

Have the authors analysed the effectiveness of the guide based on whether the only mutation within the guide would result in the generation of a STOP versus those that offer multiple mutation options?

This is an excellent suggestion and we added that this should be measured and tested in future studies. The expectation would be that guides with fewer cytosine bases within the editing window would result in higher rates of successful STOP codon transformations. If true, our guide ranking could be further modified to prioritize guides that have as few as possible cytosine bases, minimizing the possibilities for non-STOP codon mutations to become dominant (526 – 531).

In figure S11A could the authors highlight the exact sequence of the Lmajor ssDBD they used to generate the modified plasmid?

We have highlighted the amino acid sequence taken for generating the *Leishmania* optimized base editor. As requested by *eLife*, we have also updated the naming of the figure. Figure S11A is now named Figure 7—figure supplement 1.

The authors could include a discussion of the RNAi-based approaches for L. braziliensis in comparison to other methodologies.

We have revised our manuscript to include in our introduction (line 36 – 73) and discussion (line 429 – 468) a better comparison of all potential tools for genome-wide screening in *Leishmania*, including RNAi, bar-seq and base editing screening. We highlight why we think that base editing has unique advantages.